# PoinnCARE: Hyperbolic Multi-Modal Learning for Enzyme Classification

**Kun Xie[1]\***, **Peng Zhou[2]**, **Xingyi Zhang[3]**, **Wei Liu[1]**, **Peilin Zhao[4]†**, **Sibo Wang[5]†**, **Biaobin Jiang[1]†**

[1]Tencent, AI for Life Sciences Lab, Shenzhen, China;
[2]Hunan University, Changsha, China;
[3]MBZUAI, Masdar City, Abu Dhabi, UAE;
[4]Shanghai Jiao Tong University, Shanghai, China;
[5]The Chinese University of Hong Kong, Hong Kong

gennyxie@tencent.com, zhoup1366@foxmail.com,
xingyi.zhang@mbzuai.ac.ae, topliu@tencent.com,
peilinzhao@sjtu.edu.cn, swang@se.cuhk.edu.hk, brunojiang@tencent.com

## Abstract

Enzyme Commission (EC) number prediction is vital for elucidating enzyme functions and advancing biotechnology applications. However, current methods struggle to capture the hierarchical relationships among enzymes and often overlook critical structural and active site features. To bridge this gap, we introduce PoinnCARE[1], a novel framework that jointly encodes and aligns multi-modal data from enzyme sequences, structures, and active sites in hyperbolic space. By integrating graph diffusion and alignment techniques, PoinnCARE mitigates data sparsity and enriches functional representations, while hyperbolic embedding preserves the intrinsic hierarchy of the EC system with theoretical guarantees in low-dimensional spaces. Extensive experiments on four datasets from the CARE benchmark demonstrate that PoinnCARE consistently and significantly outperforms state-of-the-art methods in EC number prediction.

## 1 Introduction

Enzymes are fundamental biological catalysts that drive nearly all biochemical reactions essential for life (Berg, 2022; van Beilen and Li, 2002), and they underpin a wide range of industrial applications, including pharmaceutical synthesis (Karan et al., 2012; Nandanwar et al., 2020), food processing (Kumar et al., 2024; Kumari et al., 2021), and environmental cleaning (Gupta et al., 2002; Kumari et al., 2019). Central to understanding and harnessing enzyme function is the *Enzyme Commission (EC)* number system (Kraut, 1988; Copeland, 2023), which hierarchically classifies enzymes based on the chemical reactions they catalyze. Each EC number is a four-digit code, progressing from broad functional classes (1st digit) to highly specific activities (4th digit). For example, as illustrated in Fig. 1, the enzyme with EC number 3.1.21.1 catalyzes the hydrolysis of DNA, producing fragments with defined chemical groups at their termini. Accurate EC number prediction not only facilitates the annotation of newly discovered proteins but also enables the exploration of the vast and largely uncharacterized protein universe.

Despite recent advances (Yang et al., 2024a; Yu et al., 2023), existing computational approaches for EC number prediction face two major limitations. First, most methods either ignore or inadequately model the intrinsic hierarchical structure of the EC taxonomy, typically representing enzymes in Euclidean space (Li et al., 2018; Ryu et al., 2019; Sanderson et al., 2023). As shown in Fig. 2 (left), the EC system forms a tree-like hierarchy, which is theoretically difficult to embed in Euclidean space without significant distortion or high-dimensional overhead, limiting prediction accuracy. Second, current methods predominantly rely on sequence alignment (Stephen, 1990), overlooking critical structural and active site information that fundamentally determines enzymatic specificity and

---

\*Work done while the author was a PhD student at CUHK and a research intern at Tencent.
†Corresponding authors.
[1]Code is available at: https://github.com/kkkkk001/PoinnCARE

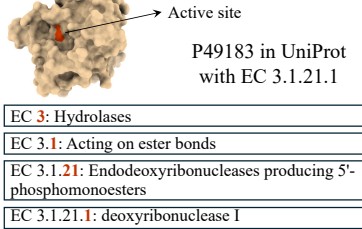

Figure 2: The tree structure of the EC system (left) and the catalytic mechanism of enzymes (right).

function. As depicted in Fig. 2 (right), enzyme catalysis depends on the precise three-dimensional arrangement of active site residues, which govern substrate binding and reaction specificity (Riziotis et al., 2025). Ignoring these modalities obscures the full complexity of enzyme function.

To address these challenges, we propose **PoinnCARE**[2], a framework that integrates multi-modal information from protein sequences, structures, and active sites, and projects them into hyperbolic space. We augment the CARE benchmark (Yang et al., 2024a) with comprehensive structural and active site annotations. While most enzymes can be assigned structures via experimental data or AlphaFold2/ESMFold predictions, experimentally validated active site annotations are available for only a small subset, leading to modality imbalance. To overcome this, PoinnCARE constructs pairwise similarity graphs for structure and active site modalities, leveraging intra-modality graph diffusion and inter-modality dual-graph alignment to alleviate annotation sparsity and bridge modality gaps. These graph representations are then projected into hyperbolic space to preserve the hierarchical relationships of the EC system. Our analysis shows that hyperbolic embeddings represent tree-like structures with lower distortion and in fewer dimensions than Euclidean approaches, while yielding better accuracy. Finally, comprehensive experiments on four test sets in the CARE benchmark (Yang et al., 2024a) show that PoinnCARE consistently outperforms state-of-the-art methods.

Active site

P49183 in UniProt
with EC 3.1.21.1

| EC **3**: Hydrolases |
| EC 3.**1**: Acting on ester bonds |
| EC 3.1.**21**: Endodeoxyribonucleases producing 5'-phosphomonoesters |
| EC 3.1.21.**1**: deoxyribonuclease I |

Figure 1: An example enzyme.

In summary, the main contributions of our work include:

- We augment the CARE benchmark with comprehensive structural and active site information, enabling richer multi-modal learning for enzyme function prediction.
- We construct similarity graphs and use graph diffusion to address annotation sparsity and enhance functional representation.
- We introduce a hyperbolic multi-modality encoding and alignment mechanism, which preserves the hierarchical relationships of the EC system with low distortion.
- We demonstrate, through extensive experiments, that PoinnCARE achieves state-of-the-art performance on four challenging EC number prediction benchmarks.

## 2 RELATED WORK

**Enzyme function prediction.** Historically, sequence similarity has been the foundation for protein function annotation (Finn et al., 2015), with BLAST (Stephen, 1990) serving as a primary tool for similarity searches. With the advancement of machine learning (ML), several methods have been proposed to leverage traditional ML technologies, such as SVM (Chang and Lin, 2011), CNN (Li et al., 2021), and ResNet (He et al., 2015), to enhance the accuracy of enzyme function annotation (Li et al., 2018; Ryu et al., 2019; Dalkiran et al., 2018; Sanderson et al., 2023). The contrastive framework was first utilized to enhance enzyme function prediction performance in CLEAN (Yu et al., 2023). Specifically, a triplet margin loss was employed to minimize distances between positive samples while maximizing distances between negative samples. Building on this simple yet powerful framework, CLEAN-Concat (Yang et al., 2024c) integrated structural information by using ResNet (He et al., 2015) to encode protein contact maps. Subsequently, several methods were introduced to augment this contrastive paradigm, including HiFi-NN (Ayres et al., 2023), FEDKEA (Zheng et al., 2024),

---

[2]Multi-modal learning with Poincaré model-based hyperbolic graph neural networks for enzyme function prediction on CARE (Yang et al., 2024a) benchmark.

EnzHier (Duan et al., 2024), and Yim et al.'s approach (Yim et al., 2024), by improving positive and negative sampling strategies based on the hierarchical characteristics of EC numbers. Recent approaches have expanded beyond protein sequences to improve the enzyme function classification results. To be specific, Top-EC (van der Weg et al., 2025) integrates enzyme structure information with a 3D graph neural network, learning from an interplay between biochemical features and local shape-dependent features. ProteinF3S (Yuan et al., 2025) consolidates sequence, structure, and surface information through a two-phase fusion strategy. Despite these advances, existing methods fail to comprehensively utilize critical information that directly determines the catalytic functions, such as active sites. These limitations motivate us to explore a novel approach incorporating multi-modal information for enhanced EC number prediction.

**Hyperbolic representation learning.** Hyperbolic geometry, characterized by negative curvature, exhibits exponential volume growth that naturally accommodates hierarchical structure and entailment relations, and thus has been successfully adopted across various domains, including computer vision Liu et al. (2020); Desai et al. (2023), natural language processing Xiong et al. (2022); Yang et al. (2024b), and graph-based tasks Sun et al. (2021); Bai et al. (2023).

In terms of the biological domain, many studies have extensively validated the superiority of hyperbolic geometry in capturing the latent hierarchical structures of biological data, successfully applying it to Gene Ontology representation Kim et al. (2021), cell lineage inference Tian et al. (2023), genomic sequence modeling Khan et al. (2025), taxonomic classification Gong et al. (2025), and protein-ligand binding Wang et al. (2025). These works collectively demonstrate that hyperbolic embeddings offer a more geometry-aware inductive bias than Euclidean approaches for modeling complex evolutionary and functional relationships. However, despite the evident hierarchical structure inherent in EC numbers, existing computational approaches heavily rely on traditional Euclidean space representations for EC number prediction, highlighting a significant gap in the field and inspiring our effort to embed enzymes, along with the rich associated information, within hyperbolic space.

## 3 PRELIMINARIES

### 3.1 PROBLEM FORMULATION

Let $\mathcal{Y}$ denote the set of EC numbers, where each element $y_i^l \in \mathcal{Y}$ represents the $i$-th EC number at hierarchical level $l \in \{0, 1, 2, 3, 4\}$. Here, $y^0$ serves as a virtual root node, acting as the common ancestor of all EC numbers. To capture hierarchical relationships among EC numbers, we establish edges $(y_i^l, y_j^{l+1})$ between nodes that share the same prefix, representing parent-child relationships. The virtual root $y^0$ connects to all first-level EC numbers $y_i^1$. With edge set $\mathcal{E}^{(t)}$, the hierarchical structure of the EC number system forms a tree $T = (\mathcal{Y}, \mathcal{E}^{(t)})$, as illustrated in Fig. 2 (left).

For any enzyme $x \in \mathcal{X}$, we represent its multi-modal information as a tuple $(q_x, s_x, a_x)$, corresponding to sequence, structure, and active site features, respectively. EC number prediction aims to learn a classifier $f(\cdot) : \mathcal{X} \rightarrow \mathcal{Y}$ mapping enzymes to their corresponding EC numbers. Importantly, since individual enzymes can catalyze multiple reactions associated with different EC numbers, this prediction task is inherently a *multi-class, multi-label* classification problem.

### 3.2 HYPERBOLIC SPACE

Hyperbolic space represents a class of Riemannian manifolds characterized by its constant negative sectional curvature (Grigor'yan and Noguchi, 1998; Dong et al., 2025). Following previous studies (Ganea et al., 2018; Yue et al., 2023; Zhang et al., 2021c), we elaborate on our method based on the Poincaré ball model. Specifically, an $n$-dimension Poincaré ball model with a constant negative curvature $\kappa(\kappa < 0)$ can be denoted as $(\mathcal{B}_\kappa^n, g_x^\kappa)$, where $\mathcal{B}_\kappa^n = \{x \in \mathbb{R}^d | \|x\|^2 < -1/\kappa\}$ represents an open ball, $g_x^\kappa = 4/(1 - \kappa\|x\|^2)^2 I$ is the Riemannian metric tensor, and $\|\cdot\|$ denotes the Euclidean norm. Equipped with this metric tensor, the induced distance between $u, v \in \mathcal{B}_\kappa^b$ is denoted as: $d(u, v) = \frac{1}{\sqrt{|\kappa|}} \text{arcosh}(1 - \frac{2\kappa\|u-v\|^2}{(1+\kappa\|u\|^2)(1+\kappa\|v\|^2)})$, which changes smoothly w.r.t. the positions of $u$ and $v$. This locality property of the hyperbolic distance is key for embedding hierarchical topologies. More details are provided in Appendix A.1.

### 3.3 TREE-LIKE RELATIONSHIPS AMONG ENZYMES

Unlike conventional flat classification with independent categories, the tree-structured EC system naturally endows enzymes with intrinsic hierarchical relationships. By connecting each enzyme to its corresponding EC number node $y_i^4 \in \mathcal{Y}$, these inherent hierarchical relationships can be quantitatively characterized through Gromov's $\delta$-hyperbolicity (Bridson and Haefliger, 2013; Gromov, 1987), as shown in Table 1. The EC system topology exhibits strong hyperbolic characteristics with a $\delta$ value close to zero, in contrast to random topologies. The computation of $\delta$-hyperbolicity is detailed in Appendix A.2.

Embedding this tree-like structure into Euclidean space poses fundamental challenges. In a tree structure, the number of nodes grows *exponentially* with depth, while the volume of an $n$-dimensional Euclidean ball only grows proportionally to the $n$-*th power* of its radius. This inherent mismatch between growth rates implies that accurate tree embedding in Euclidean space necessitates high dimensions, with limited dimensions resulting in significant distortion. In contrast, the volume of a ball in hyperbolic space grows *exponentially* with its radius,

Table 1: $\delta$-hyperbolicity.

|              | EC   | Random |
| ------------ | ---- | ------ |
| Training set | 0.01 | 0.92   |
| Test set     | 0.00 | 0.73   |

offering a natural geometric framework for embedding tree structures with faithful embeddings. An illustrative example is presented in Appendix A.3. The following theorem formally establishes this fundamental difference:

**Theorem 1** *Let $T$ be a tree with $n$ nodes and $d_T$ be the associated tree distance. Then:*

- *$(T, d_T)$ can be embedded in $O(\log n)$-dimensional Euclidean space with $O(\log n)$ distortion (Bourgain, 1985).*

- *$(T, d_T)$ can be embedded in hyperbolic space with dimension $\geq 2$ with $1 + \epsilon$ distortion, where $\epsilon$ can be arbitrarily small (Sarkar, 2011).*

## 4 METHOD: POINNCARE

In this section, we present PoinnCARE, a novel hyperbolic space-based multi-modal learning framework for EC number prediction. We first augment single-modality benchmarks with critical structural and active site information (Sec 4.1). Next, we propose a graph diffusion-enhanced topology modeling approach to capture intra-modality similarity relationships (Sec 4.2). Finally, we encode dual similarity graphs in hyperbolic space, preserving inherent hierarchical enzyme relationships while capturing cross-modal semantic correlations through inter-modality alignment (Sec 4.3).

### 4.1 MULTI-MODAL DATASET CURATION

As shown in Fig. 2 (right), the structure and active sites of enzymes are directly involved in catalytic specificity determination, and thus are crucial for understanding enzyme catalytic mechanisms (Riziotis et al., 2025). However, the existing benchmark CARE (Yang et al., 2024a) contains only sequence information, which is insufficient and indirect for enzyme function inference. Therefore, we supplement the benchmark with structure information and active site annotations, augmenting the single sequence modality to multiple modalities. Specifically, for each enzyme, we obtain the experimentally determined structures from PDB or structures predicted by AlphaFold2 (Jumper et al., 2021)/ESMFold (Lin et al., 2022). Active site annotations are obtained from UniProt (Consortium, 2024), which specify the residues directly involved in catalysis. Detailed dataset statistics are provided in Appendix C.1. Based on supplemented information, EC numbers for query enzymes can be inferred from sequence/structural/active site information through similarity search algorithms or ML/DL-based classifiers. However, the scarcity of reliable active site annotations in UniProt leads to incomplete modality information, introducing a notable gap between structural and active site modalities, which further intensifies the difficulty of multi-modal learning (Wang et al., 2024b). To address these challenges, we develop a dual similarity graph encoding framework that mitigates data sparsity from both intra-modality and inter-modality perspectives.

## 4.2 GRAPH-BASED INTRA-MODALITY RELATIONSHIP MODELING

In this section, we capture pairwise similarity under structural and active site modalities and construct two independent similarity graphs. We then employ graph diffusion operations to mitigate data sparsity by incorporating both direct and indirect connections within the graphs.

**Similarity graph under structure modality.** We employ Foldseek (Van Kempen et al., 2024) to extract pairwise structural similarity. Specifically, Fold-seek discretizes structures in continuous space and reduces 3D structure comparison to 1D sequence comparison through a VQ-VAE (Van Den Oord et al., 2017). We denote the structural similarity returned by Foldseek as $simi_s(x_i, x_j) = f_{Foldseek}(s_{x_i}, s_{x_j})$, where $x_i, x_j \in \mathcal{D}$ denoting the enzymes from the dataset, and $s_x$ represents the corresponding structure. Based on the score $simi_s(\cdot, \cdot)$, we construct a similarity graph under the structure modality, denoted as $G^{(s)} = (\mathcal{D}, \mathcal{E}^{(s)})$. An edge $(x_i, x_j)$ is included in the edge set $\mathcal{E}^{(s)}$ if $simi_s(x_i, x_j) > \delta^s$, where $\delta^s$ is a predefined threshold. Graph construction details and statistics are provided in Appendix B.

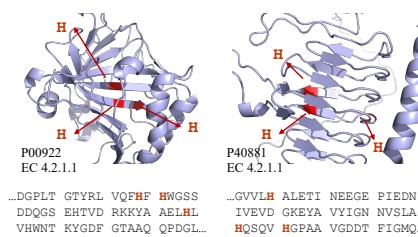

Figure 3: Active site illustration.

**Similarity graph under active site modality.** We derive the enzyme similarity from the perspective of active sites using Folddisco (Kim et al., 2025), an inverted-index-based method for fast structural motif detection within databases. Given the geometry and amino acid types of the active sites of $x_i$, Folddisco first identifies whether a similar motif exists in $x_j$. If a local structure in $x_j$ is identified, Folddisco computes the similarity between the query active sites and the identified one, denoted as $simi_a(x_i, x_j) = f_{Folddisco}\big((q_{x_i}, s_{x_i}, a_{x_i}), (q_{x_j}, s_{x_j})\big)$. We denote the similarity graph under active site modality as $G^{(a)} = (\mathcal{D}, \mathcal{E}^{(a)})$, where edge set $\mathcal{E}^{(a)}$ includes $(x_i, x_j)$ if there is a local motif in $x_j$ sharing high similarity with the active sites of $x_i$, i.e., $simi_a(x_i, x_j) > \delta^a$.

Active site information provides complementary insights beyond sequence and structural features. As illustrated in Fig. 3, enzymes can share identical EC numbers and active sites while exhibiting distinct sequence and structural patterns. This phenomenon arises because active site residues are typically scattered and discontinuous in the sequence (Hu et al., 2024), and their local structural features may deviate from the global protein structure distribution (Riziotis et al., 2025).

*Remark.* When learning over these similarity graphs, we follow the inductive learning paradigm (Hamilton et al., 2017), strictly ensuring that *only* training enzymes and relationships among training enzymes are visible during the training phase, as further detailed in Appendix D.4.

**Graph diffusion.** Let $\boldsymbol{A}_s, \boldsymbol{A}_a$ be the adjacency matrices of the structural and active site similarity graphs, respectively. We augment the topology of these two graphs by aggregating information from multi-hop neighbors through a graph diffusion operation:

$$\boldsymbol{A}'_s = \sum_{k=0}^{\infty} w_k^s \boldsymbol{P}_s^k, \quad \boldsymbol{A}'_a = \sum_{k=0}^{\infty} w_k^a \boldsymbol{P}_a^k. \tag{1}$$

Specifically, $\boldsymbol{P}_a$ is the transition matrix of the active site similarity graph and $w_k^a$ is the weighting coefficient at $k$-th hop satisfying $\sum_{k=0}^{\infty} w_k^a = 1$. Graph diffusion can be instantiated into different formulations (Gasteiger et al., 2018; 2019; Kipf and Welling, 2016; Wu et al., 2019). Let $\boldsymbol{D}_a$ be the degree matrix of $A_a$. In this work, we set $\boldsymbol{P}_a = \boldsymbol{D}_a^{-1}\boldsymbol{A}_a$, $w_k^a = \alpha_a(1 - \alpha_a)^k$, and restrict the sum to a finite number $L_a$, yielding the personalized PageRank distribution (Wang et al., 2017; Zhang et al., 2021b; 2024; Xie et al., 2024; 2025; Hou et al., 2021). Similar notations are adopted for the structure similarity graph.

The enhanced distributions $\boldsymbol{A}'_s$ and $\boldsymbol{A}'_a$ can be viewed as weighted and directed graphs, with edge weights reflecting the connection strength between node pairs, taking into account both their direct connections and indirect connections through multi-hop neighbors. Following (Gasteiger et al., 2019), we preserve these weights and use the resulting weighted graphs for subsequent hyperbolic encoding and alignment.

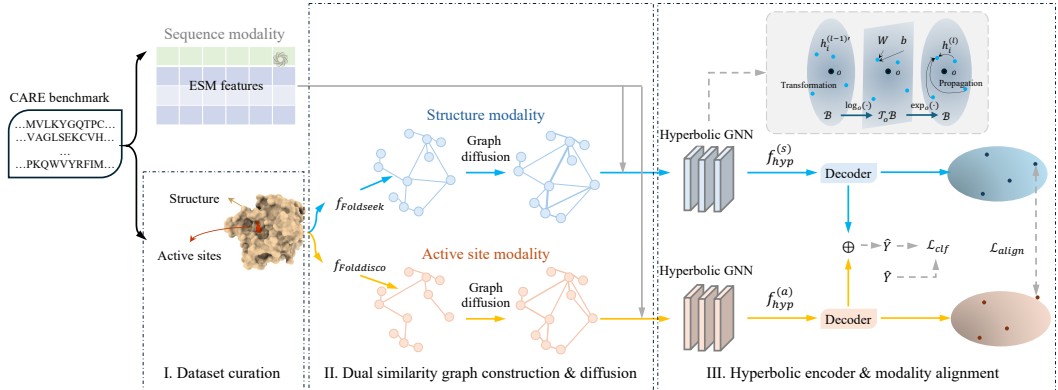

Figure 4: Overview of PoinnCARE framework: I. Curation of a multi-modal dataset by augmenting existing benchmarks with structure and active site information; II. Capturing intra-modality relationships with graph modeling and graph diffusion; III. Encoding enzyme similarity in hyperbolic space using dual hyperbolic GNNs to capture inter-modality relationships with modality alignment.

## 4.3 MULTI-MODAL LEARNING AND ALIGNING IN HYPERBOLIC SPACE

Subsequently, we encode the augmented similarity graphs with two independent GNNs in hyperbolic space to preserve the intrinsic hierarchical topologies, guided by cross-modality alignment loss to capture the inter-modality invariance.

Standard GNN layer updates node representations in three sequential stages: linear transformation, neighbor aggregation, and non-linear activation (Liu et al., 2023; Wu et al., 2019; Zhang et al., 2023). However, these foundational operations are not readily applicable in hyperbolic space. To address this issue, a local Euclidean approximation within hyperbolic space is commonly adopted as a compromise (Ganea et al., 2018; Shimizu et al., 2020).

**Definition 1 (Tangent Space)** *For a point $x \in \mathcal{B}_\kappa^n$ in hyperbolic space, its tangent space $\mathcal{T}_x \mathcal{B}_\kappa^n$ provides a first-order approximation of $\mathcal{B}_\kappa^n$ at $x$ and is isomorphic to Euclidean space.*

The exponential map operation $\exp_x : \mathcal{T}_x \mathcal{B}_\kappa^n \to \mathcal{B}_\kappa^n$ projects vectors from the tangent space back to the hyperbolic space, while the logarithmic map $\log_x : \mathcal{B}_\kappa^n \to \mathcal{T}_x \mathcal{B}_\kappa^n$ performs the inverse operation. Furthermore, the parallel transport $\mathrm{PT}_{x \to y} : \mathcal{T}_x \mathcal{B}_\kappa^n \to \mathcal{T}_y \mathcal{B}_\kappa^n$ defines a way of transporting the local geometry along smooth curves that preserve the metric tensors. The detailed mathematical formulations are provided in Appendix A.4.

Based on these definitions, we update node embeddings in hyperbolic space by first projecting them onto the tangent space, performing standard operations (linear transformation and neighbor aggregation), and finally mapping results back to hyperbolic space (Chami et al., 2019; Ganea et al., 2018; Shimizu et al., 2020; Zhang et al., 2021c). For simplicity, the tangent space of the origin node is selected to perform standard operations. Specifically, the matrix multiplication with $\boldsymbol{W}$ and the bias translation with $b$ in hyperbolic space can be formulated as:

$$\boldsymbol{W} \otimes \boldsymbol{x} = \exp_o \left( \boldsymbol{W} \log_o(\boldsymbol{x}) \right), \quad \boldsymbol{x} \oplus \boldsymbol{b} = \exp_x \left( \mathrm{PT}_{o \to x}(\boldsymbol{b}) \right). \tag{2}$$

Then, the message passing procedure at the $l$-th layer is:

$$\boldsymbol{h}_i^{(l)'} = f_{trans} \left( \boldsymbol{h}_i^{(l)} \right) = \left( \boldsymbol{W} \otimes \boldsymbol{h}_j^{(l)} \right) \oplus \boldsymbol{b}, \tag{3}$$

$$\boldsymbol{h}_i^{(l+1)} = \delta \left( f_{agg}(\boldsymbol{h}_i^{(l)'}) \right) = \delta \left( \exp_o \left( \sum_{j \in N(i)} a_{ij} \log_o \left( \boldsymbol{h}_j^{(l)'} \right) \right) \right), \tag{4}$$

where $N(i)$ denotes the neighbor set of node $i$, and $\delta(\cdot)$ is a non-linear activation function, such as ReLU. We set the weight $a_{ij}$ according to the normalized Laplacian matrix (Kipf and Welling, 2016; Liu et al., 2019). The input features $h_i^{(0)}$ are initialized using a Protein Language Model (PLM), such as ESM (Lin et al., 2022) used in (Yang et al., 2024c; Yu et al., 2023). Compared to Euclidean GNNs,

the hyperbolic overhead introduces an additional computational cost of $O(nd)$, increasing the total computational complexity from $O(mnd)$ in a standard model to $O(mnd + nd)$ in the hyperbolic GNN, where $n$, $m$, and $d$ denote the number of nodes, edges, and feature dimensions, respectively.

We separately encode structural and the active site information with two independent hyperbolic GNNs. Specifically, we feed the adjacency matrices augmented by graph diffusion, $A'_s$ and $A'_a$, along with PLM embeddings $H^{(0)}$ into the dual hyperbolic GNNs:

$$\boldsymbol{H}_{(s)} = f_{hyp}^{(s)}(\boldsymbol{A}'_s, \boldsymbol{H}^{(0)}), \quad \boldsymbol{H}_{(a)} = f_{hyp}^{(a)}(\boldsymbol{A}'_a, \boldsymbol{H}^{(0)}). \tag{5}$$

We then align the representations under two modalities by minimizing the divergence between structural and active site embeddings of the same enzyme (Zhang et al., 2021a):

$$\mathcal{L}_{align} = \|\boldsymbol{H}_{(s)} - \boldsymbol{H}_{(a)}\|_F^2 + w_d(\|\boldsymbol{I} - \boldsymbol{H}_{(s)}^\top \boldsymbol{H}_{(s)}\|_F^2 + \|\boldsymbol{I} - \boldsymbol{H}_{(a)}^\top \boldsymbol{H}_{(a)}\|_F^2). \tag{6}$$

The first term of $\mathcal{L}_{align}$ maximizes the correlation between representations from two modalities and captures the invariance between different views. The subsequent decorrelation terms prevent learning degenerated embeddings (Liu et al., 2023), with $w_d$ controlling the weight of decorrelation.

The representations from both modalities are then combined to derive the EC number prediction $\hat{Y}$, which is formulated as a weighted sum of the modality-specific predictions, governed by the trade-off parameters $\beta_s$ and $\beta_a$ (He et al., 2023):

$$\hat{Y} = \beta_s \cdot f_{clf}^{(s)}(\boldsymbol{H}_{(s)}) + \beta_a \cdot f_{clf}^{(a)}(\boldsymbol{H}_{(a)}). \tag{7}$$

Finally, the overall model is optimized by minimizing a joint objective function that balances the alignment loss and the cross-entropy loss: $\mathcal{L} = \mathcal{L}_{align} + \gamma \mathcal{L}_{ce}$.

## 5 EXPERIMENTS

### 5.1 EXPERIMENTAL SETTINGS

**Dataset.** We evaluate PoinnCARE on the standardized enzyme function benchmark CARE (Yang et al., 2024a) curated from Swiss-Prot (Consortium, 2024), comprising enzymes with validated four-digit EC number annotations. To rigorously assess the generalizability of a model to unseen proteins, CARE defines four distinct test sets, each presenting unique challenges:

- *<30% Identity* test set: All enzymes in the test set share less than 30% sequence identity with enzymes in the training set, ensuring stringent low-homology testing conditions (Rost, 1999).
- *30-50% Identity* test set: Enzymes in this test set share sequence identity between 30% and 50% with those in the training set, representing an intermediate homology zone.
- *Previously Misclassified (Price)* test set: A collection of enzymes that were initially misannotated in established databases such as KEGG by automated annotation methods, but were subsequently experimentally validated and correctly reclassified by Price et al. (Price et al., 2018).
- *Promiscuous* test set: A collection of enzymes capable of catalyzing multiple distinct reactions that are classified under different EC numbers. In this dataset, a single enzyme can be associated with up to 9 different EC numbers.

All four test sets share a common training set. Statistics of training and test sets are presented in Table 5. Following the recommendation in the CARE benchmark, we use 50% sequence clustering of the training set to increase the diversity. We follow the **inductive setting** (Hamilton et al., 2017), ensuring that only training enzymes are accessible during training, while test enzymes are withheld until inference. Appendix D.4 provides more explanations and the performance comparison.

**Baselines.** We compare PoinnCARE with 12 SOTA competitors belonging to four categories:

- Similarity search algorithms: BLASTp (Stephen, 1990), Foldseek (Van Kempen et al., 2024), Folddisco (Kim et al., 2025);
- Contrastive learning methods: CLEAN (Yu et al., 2023), CLEAN-Concat (Yang et al., 2024c);
- PLMs for general proteins: ESM-2 (Lin et al., 2022), ESM-c (ESM Team, 2024), ProtT5 (Elnaggar et al., 2020), ProtBert (Elnaggar et al., 2020), S-PLM (Wang et al., 2024a);

Table 2: Performance regarding accuracy on <30% Identity and 30-50% Identity test sets. The best and second-best results are shown in bold and underlined, respectively.

| | <30% Identity | | | | 30-50% Identity | | | | Avg. rank |
| | Level 1 (x.-.-.-) | Level 2 (x.x.-.-) | Level 3 (x.x.x.-) | Level 4 (x.x.x.x) | Level 1 (x.-.-.-) | Level 2 (x.x.-.-) | Level 3 (x.x.x.-) | Level 4 (x.x.x.x) | |
|---|---|---|---|---|---|---|---|---|---|
| Random* | 0.194 | 0.032 | 0.012 | 0.000 | 0.225 | 0.036 | 0.007 | 0.000 | 13.38 |
| BLASTp | 0.697 | 0.590 | 0.569 | 0.475 | 0.923 | 0.879 | 0.850 | 0.773 | 7.00 |
| Foldseek | 0.815 | 0.722 | 0.667 | 0.544 | 0.932 | 0.880 | 0.841 | 0.755 | 4.13 |
| Folddisco | 0.756 | 0.600 | 0.511 | 0.378 | 0.798 | 0.755 | 0.723 | 0.564 | 9.88 |
| CLEAN | 0.806 | 0.729 | 0.678 | 0.535 | 0.946 | 0.905 | 0.870 | 0.798 | 2.50 |
| CLEAN-Concat | 0.810 | 0.704 | 0.646 | 0.507 | 0.946 | 0.893 | 0.859 | 0.777 | 3.63 |
| ESM-2 | 0.783 | 0.695 | 0.643 | 0.518 | 0.944 | 0.895 | 0.856 | 0.781 | 4.13 |
| ESM-c | 0.691 | 0.574 | 0.527 | 0.436 | 0.911 | 0.848 | 0.808 | 0.745 | 9.25 |
| ProtT5 | 0.755 | 0.649 | 0.604 | 0.492 | 0.929 | 0.873 | 0.833 | 0.765 | 6.38 |
| ProtBert | 0.672 | 0.546 | 0.502 | 0.410 | 0.874 | 0.804 | 0.767 | 0.709 | 10.38 |
| S-PLM | 0.751 | 0.637 | 0.582 | 0.470 | 0.921 | 0.861 | 0.823 | 0.752 | 7.75 |
| ChatGPT* | 0.278 | 0.016 | 0.000 | 0.000 | 0.336 | 0.030 | 0.014 | 0.000 | 13.38 |
| Pika* | 0.616 | 0.461 | 0.377 | 0.206 | 0.738 | 0.600 | 0.502 | 0.377 | 12.00 |
| PoinnCARE | **0.900** | **0.827** | **0.779** | **0.648** | **0.961** | **0.926** | **0.887** | **0.822** | 1.00 |

Table 3: Performance regarding accuracy on Price and Promiscuous test sets. The best and second-best results are shown in bold and underlined, respectively.

| | Previously Misclassidied (Price) | | | | Promiscuous | | | | Avg. rank |
| | Level 1 (x.-.-.-) | Level 2 (x.x.-.-) | Level 3 (x.x.x.-) | Level 4 (x.x.x.x) | Level 1 (x.-.-.-) | Level 2 (x.x.-.-) | Level 3 (x.x.x.-) | Level 4 (x.x.x.x) | |
|---|---|---|---|---|---|---|---|---|---|
| Random* | 0.223 | 0.047 | 0.007 | 0.000 | 0.411 | 0.090 | 0.041 | 0.005 | 12.88 |
| BLASTp | 0.824 | 0.811 | 0.710 | 0.341 | 0.843 | 0.784 | 0.733 | 0.682 | 5.63 |
| Foldseek | 0.939 | 0.878 | 0.797 | 0.314 | 0.769 | 0.689 | 0.638 | 0.561 | 6.38 |
| Folddisco | 0.000 | 0.000 | 0.000 | 0.000 | 0.656 | 0.526 | 0.484 | 0.318 | 12.25 |
| CLEAN | 0.858 | 0.797 | 0.696 | 0.280 | 0.873 | 0.816 | 0.768 | 0.691 | 5.00 |
| CLEAN-Concat | 0.905 | 0.872 | 0.770 | 0.348 | 0.874 | 0.813 | 0.776 | 0.659 | 3.13 |
| ESM-2 | 0.918 | 0.849 | 0.762 | 0.362 | 0.861 | 0.780 | 0.724 | 0.629 | 4.00 |
| ESM-c | 0.791 | 0.726 | 0.666 | 0.343 | 0.818 | 0.730 | 0.672 | 0.579 | 8.13 |
| ProtT5 | 0.895 | 0.827 | 0.761 | **0.380** | 0.843 | 0.754 | 0.694 | 0.596 | 5.38 |
| ProtBert | 0.678 | 0.554 | 0.515 | 0.200 | 0.814 | 0.707 | 0.636 | 0.549 | 10.00 |
| S-PLM | 0.872 | 0.789 | 0.719 | 0.238 | 0.848 | 0.761 | 0.709 | 0.606 | 6.50 |
| ChatGPT* | 0.365 | 0.169 | 0.088 | 0.000 | 0.196 | 0.055 | 0.036 | 0.002 | 13.00 |
| Pika* | 0.824 | 0.649 | 0.507 | 0.041 | 0.618 | 0.473 | 0.372 | 0.164 | 11.00 |
| PoinnCARE | **0.955** | **0.909** | **0.827** | 0.349 | **0.911** | **0.871** | **0.849** | **0.785** | 1.25 |

- LLMs for general protein-related question answering: GPT-4o-mini (Hurst et al., 2024), Pika (Carrami and Sharifzadeh, 2024).

A brief introduction to the baseline methods is provided in Appendix C.2, with Table 6 summarizing the modalities employed by each baseline method. The results of LLMs and the random baseline are directly adopted from the CARE benchmark and marked with an asterisk *.

**Metrics.** For evaluation, we adopt two sets of metrics: the *accuracy* score as defined in the CARE benchmark (Yang et al., 2024a), and the *precision, recall, and $F_1$* scores adopted in CLEAN (Yu et al., 2023). Following the evaluation protocol established in (Yang et al., 2024a; Yu et al., 2023; Yang et al., 2024c), we evaluate the classification performance at all four EC number levels. Level 1 evaluation solely examines the correctness of the first digit, while level 4 evaluation requires the accurate prediction of all four digits in the EC number. Further implementation details are provided in Appendix C.3

## 5.2 CLASSIFICATION RESULTS

Tables 2 and 3 present a comprehensive comparison between PoinnCARE and 12 state-of-the-art methods on the CARE benchmark in terms of accuracy. Across all four evaluation levels and test sets, PoinnCARE consistently achieves the highest accuracy in nearly all cases, demonstrating robust and superior performance. Compared to the strongest baseline CLEAN, PoinnCARE achieves

substantial improvements of 10.4% and 2.4% in level 4 accuracy on <30% Identity and 30-50% Identity test sets, respectively, demonstrating its exceptional capability in EC number prediction under low sequence similarity conditions. On the challenging Price test set, PoinnCARE outperforms the second-best method, with accuracy improvements of 1.7%, 3.1%, and 3.0% at levels 1 through 3, respectively. Furthermore, for enzymes with promiscuous functions, PoinnCARE surpasses the second-best competitor CLEAN by a significant margin of 9.4% in level 4 accuracy. In terms of precision, recall, and $F_1$ score, PoinnCARE also demonstrates a similar outstanding performance, with detailed results presented in Appendix D.1. The standard deviation values are in Appendix D.2.

As illustrated in the left sub-figure of Fig 2, the classification space expands from 7 main classes at level 1 to over 4,900 distinct EC numbers at level 4, indicating a notable increase in classification complexity across levels. Nevertheless, PoinnCARE consistently outperforms existing methods across all four levels, demonstrating remarkable robustness. Among the 12 baseline methods, Foldseek, S-PLM, and CLEAN-Concat incorporate structural information of enzymes. Notably, Foldseek exhibits promising performance on the <30% Identity test set, highlighting the effectiveness of structural information, especially when sequence similarity is highly limited. We also report the performance of valid Folddisco predictions. While BLASTp utilizes full sequence information containing hundreds of residues, Folddisco achieves comparable performance by leveraging only several residues in local active site motifs. This comparison demonstrates the crucial role of active sites in determining enzyme function, particularly in cases of low sequence similarity.

## 5.3 CLASSIFICATION RESULTS IN A LIMITED-DIMENSIONAL SPACE

As shown in Theorem 1, compared with Euclidean space, hyperbolic space can achieve arbitrarily low distortion even in low dimensions. Following this theoretical analysis, we evaluate the dimensional efficiency of PoinnCARE against CLEAN, the strong baseline, by systematically reducing the

embedding dimension from 512 to 32, and the results on the <30% Identity test set are presented in Figure 5. As the embedding dimension decreases from 512 to 32, CLEAN's classification accuracy at level 4 drops significantly from 0.535 to 0.354, suffering an 18.1% performance degradation. In contrast, our PoinnCARE maintains robust performance across different dimensions. Notably, even with a compact 32-dimensional representation, PoinnCARE achieves a strong accuracy of 0.597 at level 4. Appendix D.3 provides detailed results on the <30% and 30-50% Identity test sets.

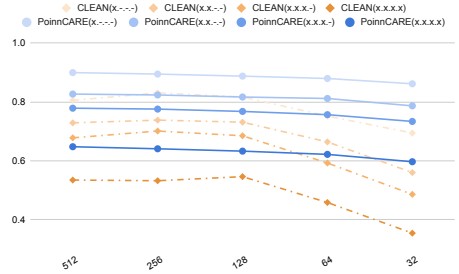

Figure 5: Varying dimensions.

## 5.4 POINNCARE AS A GENERAL FRAMEWORK

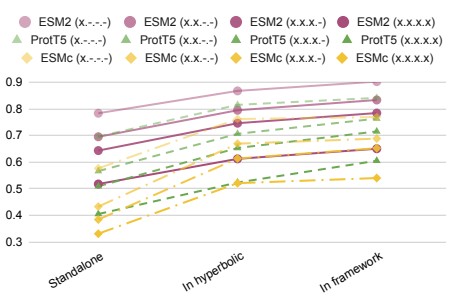

Figure 6: PoinnCARE as a framework.

In this section, we explore the generalizability of PoinnCARE by treating it as a general framework that can be combined with different sequence encoders. For ESM2 (Lin et al., 2022), ESMc (ESM Team, 2024), and ProtT5 (Heinzinger et al., 2024), we present the standalone performance, the performance in hyperbolic space, and the performance when integrated into our PoinnCARE framework in Figure 6. Compared to the standalone performance, transforming into hyperbolic space yields notable improvements in level-4 accuracy: 10.6% for ESM2, 11.8% for ProtT5, and 19.0% for ESMc. Integrating with the full PoinnCARE framework further boosts performance, achieving up to an additional 8.2% improvement in level-4 accuracy.

## 5.5 ABLATION STUDY

In this section, we conduct ablation studies in a *bottom-up manner*. Starting from a naive MLP classifier, we progressively incorporate key components to demonstrate their individual contributions, culminating in the full PoinnCARE framework. Figure 7 shows how the accuracy evolves during this process on the <30% Identity test set. First, transforming the MLP into hyperbolic space yields a substantial improvement of 9.3% in level-4 accuracy. Subsequently, the independent incorporation of the active site and the structural similarity graph each further enhances performance. Finally, integrating modality alignment to fuse both information notably boosts overall results, achiev-

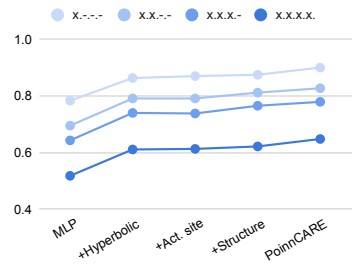

Figure 7: Ablation study.

ing a 2.6% improvement over the variant with only structural information in level-4 accuracy. A case study illustrating the complementarity between the two modalities is presented in Appendix E. Furthermore, Appendix D.4 provides a comprehensive parameter analysis, including investigations into the inductive learning setting, the curvature of the underlying hyperbolic space, and the graph diffusion settings.

## 6 CONCLUSION

In this paper, we present PoinnCARE, a hyperbolic multi-modal learning framework for enzyme function prediction. By effectively incorporating both structural and active site information through graph-based modeling and diffusion, PoinnCARE captures comprehensive enzyme characteristics beyond sequence features. Motivated by the theoretical advantages when embedding trees, we adopt hyperbolic geometry instead of traditional Euclidean space for enzyme representation learning. Then we align and fuse information from these two complementary modalities, capturing comprehensive enzyme characteristics. Extensive experiments on the CARE benchmark demonstrate that PoinnCARE consistently outperforms existing methods across various challenging scenarios.

## ACKNOWLEDGMENTS

Sibo Wang is supported by the RGC GRF grant (No. 14217322) and the "1+1+1" CUHK-CUHK(SZ)-GDST Joint Collaboration Fund.

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

# A   HYPERBOLIC GEOMETRY

## A.1   POINCARÉ BALL MODEL

As we presented before, an $n$-dimension Poincaré ball model with a constant negative curvature $\kappa(\kappa < 0)$ can be denoted as $(\mathcal{B}_\kappa^n, g_x^\kappa)$, where $\mathcal{B}_\kappa^n = \{x \in \mathbb{R}^d | \|x\|^2 < -1/\kappa\}$ is an open ball with radius $(-1/\kappa)^{1/2}$, $g_x^\kappa = 4/(1 - \kappa\|x\|^2)^2 I$ is the Riemannian metric tensor, and $\|\cdot\|$ denotes the Euclidean norm.

The metric tensor in Euclidean space is $g^E = I$. This implies that the metric tensor of Poincaré model is conformal to that in Euclidean space, meaning that the angles defined in Poincaré ball model are the same as those in Euclidean space. The conformal factor between them is $\lambda_x = 2/(1 - \kappa\|x\|^2)$, and the Poncaré model's metric tensor can be rearranged as $g_x^\kappa = \lambda_x^2 I$.

The distance between $u, v \in \mathcal{B}_\kappa^b$ is denoted as:

$$d(u, v) = \frac{1}{\sqrt{|\kappa|}}\text{arcosh}(1 - \frac{2\kappa\|u - v\|^2}{(1 + \kappa\|u\|^2)(1 + \kappa\|v\|^2)}).$$

The hyperbolic distance is determined by both the relative positions of points $u$ and $v$ ($\|u - v\|^2$ in the numerator) and their absolute positions ($\|u\|^2$ and $\|v\|^2$ in the denominator). Notably, for points approaching the boundary of the Poincaré ball (where their norms approach $-1/\kappa$), the hyperbolic distance grows significantly faster than the corresponding Euclidean distance. This fast growth of space near the boundary is particularly advantageous for embedding tree-like hierarchical structures. Leaf nodes belonging to different subtrees can be placed far apart in this near-boundary space, while maintaining their relative distances within the same subtree, thus accurately preserving the structure of the hierarchy.

## A.2   GROMOV HYPERBOLICITY

Gromov's $\delta$-hyperbolicity (Bridson and Haefliger, 2013; Gromov, 1987; Narayan and Saniee, 2011) is a notion from group theory, measuring how tree-like a graph is. Specifically, let $(G, d_G)$ denote the input graph with its associated distance function. Let $a, b, c, d$ be four vertices of the input graph, and define $S_1, S_2, S_3$ as:

$$S_1 = d_G(a, b) + d_G(d, c), \quad S_2 = d_G(a, c) + d_G(b, d), \quad S_3 = d_G(a, d) + d_G(b, c).$$

Then we can calculate $\delta(a, b, c, d)$ as the difference between largest and the second largest largest $S_.$, denoted as $M_1$ and $M_2$, respectively:

$$\delta(a, b, c, d) = M_1 - M_2.$$

We analyze the hyperbolic characteristics of graph $G$ using the mean of sampled $\delta(a, b, c, d)$ values, where $a, b, c, d$ are randomly selected nodes in $G$, to provide a statistical view of the graph's geometric properties. While the classical $\delta$-hyperbolicity is defined as the supremum of all $\delta(a, b, c, d)$, we use this mean-based metric to capture the average hyperbolic behavior of the topology.

For tree topologies, we have $\delta = 0$. A $\delta$ value that is close to zero indicates that the input structure of the graph more closely resembles a tree-like hierarchical organization. The closer the $\delta$ to zero, the more tree-like (or more hyperbolic) the given topology is.

### A.3 EMBEDDING A REGULAR TREE IN TWO-DIMENSIONAL SPACES

To demonstrate how Euclidean and hyperbolic spaces fundamentally differ in their capacity for embedding hierarchical structures, we analyze a concrete example: embedding a regular tree in two-dimensional spaces.

Consider a regular tree with branching factor $b$, where each node has exactly $b$ child nodes. The total number of nodes in an $l$-layer tree is $\sum_{i=0}^{l} b^i$, exhibiting exponential growth with respect to the depth $l$. When embedding such a tree into a geometric space, we can place the root node at the origin with leaf nodes extending outward. In Euclidean space, however, the available area grows insufficiently: a disc of radius $l$ has area $2\pi l^2$, scaling only quadratically with $l$. In contrast, the area of a disc with radius $l$ in hyperbolic space (with curvature $-1$) is $2\pi(e^l/2 + e^{-l}/2 - 1)$, providing exponential growth that matches the tree's expansion rate.

### A.4 TANGENT SPACE RELATED OPERATIONS

In this section, we provide a brief introduction to the tangent-space related operations, starting from a basic concept, Möbius addition (Ungar, 2007), that is utilized in deriving the closed-form expression of other operations.

**Möbius addition.** For $x, y \in \mathcal{B}_\kappa^n$, the Möbius addition is defined as:

$$x \oplus_\kappa y = \frac{(1 - 2\kappa\langle x, y\rangle_2 - \kappa\|y\|^2)x + (1 + \kappa\|x\|^2)y}{1 - 2\kappa\langle x, y\rangle_2 + \kappa^2\|x\|^2\|y\|^2}. \tag{8}$$

Note that when $\kappa = 0$, the Möbius addition degenerates to the Euclidean addition. Based on the Möbius addition, we can define the Möbius subtraction $x \ominus_\kappa y = x \oplus_\kappa (-y)$. Next, we present the detailed mathematical formulation of the exponential map, logarithmic map, and parallel transport (Ganea et al., 2018; Yang et al., 2023).

**Exponential map.** For $x \in \mathcal{B}_\kappa^n$ and $v \in T_x\mathcal{B}_\kappa^n, v \neq 0$, the exponential map $\exp_x : \mathcal{T}_x\mathcal{B}_\kappa^n \to \mathcal{B}_\kappa^n$ is defined as:

$$\exp_x(v) = x \oplus_\kappa \left(\tanh(\sqrt{|\kappa|}\frac{\lambda_x\|v\|}{2})\frac{v}{\sqrt{|\kappa|}\|v\|}\right), \tag{9}$$

where $\lambda_x = 2/(1 - \kappa\|x\|^2)$ is the conformal factor.

**Logarithmic map.** For $x, y \in \mathcal{B}_\kappa^n, x \neq y$, the logarithmic map $\log_x : \mathcal{B}_\kappa^n \to \mathcal{T}_x\mathcal{B}_\kappa^n$ is defined as

$$\log_x(y) = \frac{2}{\sqrt{|\kappa|}\lambda_x}\tanh^{-1}(\sqrt{|\kappa|}\| - x \oplus_\kappa y\|)\frac{-x \oplus_\kappa y}{\| - x \oplus_\kappa y\|}. \tag{10}$$

**Parallel transport.** Here we present the parallel transport $PT_{0\to x} : \mathcal{T}_0\mathcal{B}_\kappa^n \to \mathcal{T}_x\mathcal{B}_\kappa^n$ that transports a vector $v \in \mathcal{T}_0\mathcal{B}_\kappa^n$ to a another tangent space $\mathcal{T}_x\mathcal{B}_\kappa^n$:

$$PT_{0\to x}(v) = \log_x(x \oplus_\kappa \exp_0(v)) = \frac{\lambda_0}{\lambda_x}v. \tag{11}$$

## B SIMILARITY GRAPH CONSTRUCTION DETAILS

### B.1 STRUCTURE MODALITY

For the structure modality, we employ Foldseek (Van Kempen et al., 2024) to compute pairwise similarities. For enzymes $x_i, x_j \in \mathcal{D}$ with their corresponding structures $s_{x_i}$ and $s_{x_j}$, the similarity is measured by the normalized bit score:

$$simi_s(x_i, x_j) = f_{Foldseek}(s_{x_i}, s_{x_j}) = \frac{\text{bits}(s_{x_i}, s_{x_j})}{\text{bits}(s_{x_i}, s_{x_i})}$$

where $\text{bits}(s_{x_i}, s_{x_j})$ is Foldseek's default scoring metric measuring the structural alignment between $s_{x_i}$ and $s_{x_j}$. The normalization by self-alignment score ensures the similarity measure is bounded between 0 and 1. The edge set $\mathcal{E}^{(s)}$ includes an edge between enzymes $x_i$ and $x_j$ when $simi_s(x_i, x_j) > \delta^s$, where $\delta^s = 0.3$.

## B.2 ACTIVE SITE MODALITY

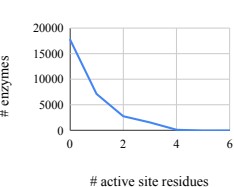

Figure 8: Distribution.

Fig. 8 illustrates the distribution of active site annotations from UniProt across our enzyme dataset, where more than half of the enzymes have no annotated active site residues. Based on the available annotations, we construct the active site modality graph using Folddisco (Kim et al., 2025) to identify similar local motifs. For enzyme $x_i$ with sequence $q_{x_i}$, structure $s_{x_i}$, and active site annotation $a_{x_i}$, Folddisco first identifies whether a similar motif exists in enzyme $x_j$. If a match is found, Folddisco returns two metrics: RMSD (Root Mean Square Deviation), measuring the geometric similarity, and IDF (Inverse Document Frequency), quantifying the rarity of the identified local structural patterns. We filter out matches with IDF $\leq 0.7$ to focus on meaningful structural patterns, then define the similarity between enzymes using the RMSD score:

$$simi_a(x_i, x_j) = f_{Folddisco}((q_{x_i}, s_{x_i}, a_{x_i}), (q_{x_j}, s_{x_j})) = \text{RMSD}(x_i, x_j)$$

The edge set $\mathcal{E}^{(a)}$ includes an edge between enzymes $x_i$ and $x_j$ when $simi_a(x_i, x_j) > \delta^a$, where $\delta^a = 0.05$.

## B.3 GRAPH STATISTICS

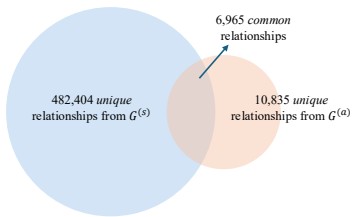

Figure 9: Edge distribution.

Table 4 presents the key statistics of both similarity graphs. The homophily ratio, defined as the proportion of edges connecting enzymes with identical EC numbers (Zhu et al., 2020), indicates that the structure modality graph $G^{(s)}$ exhibits stronger homophily than the active site modality graph $G^{(a)}$. The node set contains enzymes in both the training and test sets. Following the recommendation in the CARE benchmark, we use 50% sequence clustering of the training set to increase the set diversity. We also analyze the distribution of structural and active site similarity graphs, as shown in Fig. 9, which reveals a modest number of shared relationships, indicating that each modality captures distinct and unique information.

Table 4: Statistics of similarity graphs $G^{(s)}$ and $G^{(a)}$.

|  | # nodes | # edges | Homophily Ratio |
| --- | --- | --- | --- |
| $G^{(s)}$ | 29,531 | 489,369 | 0.808 |
| $G^{(a)}$ | 29,531 | 17,800 | 0.514 |

## C ADDITIONAL EXPERIMENTAL SETTINGS

### C.1 DATA STATISTICS

Table 5 shows the detailed statistics of the utilized benchmark CARE.

### C.2 BASELINES

We include the following SOTA methods as baselines:

- BLASTp (Stephen, 1990) is a fast sequence alignment algorithm that can search for database sequences similar to a given query sequence.

Table 5: Dataset statistics.

|  | # enzymes | # unique EC numbers |
|---|---|---|
| Training set | 184,529 | 4936 |
| <30% Identity test set | 432 | 333 |
| 30-50% Identity test set | 560 | 389 |
| Previously Misclassified (Price) test set | 148 | 56 |
| Promiscuous test set | 209 | 384 |

- Foldseek (Van Kempen et al., 2024) is a fast structure alignment algorithm. By reducing 3D structural information to 1D sequence information, Foldseek can search for database structures similar to a given query structure.
- Folddisco (Kim et al., 2025) is a fast local motif detection algorithm that can search and identify local structures from the database that are similar to a given motif.
- CLEAN (Yu et al., 2023) initializes enzyme embeddings with ESM-2 (Lin et al., 2022), and then optimizes embeddings with a triplet margin loss, where the distance between pairs with the same EC class is minimized and the distance between pairs with different EC classes is maximized.
- CLEAN-Concat (Yang et al., 2024c) improves the CLEAN method by utilizing both sequence information in ESM-2 embedding and the structure information encoded by ResNet (He et al., 2015).
- ESM-2 (Lin et al., 2022) is a protein language model trained on sequences of natural proteins with the aim of designing protein structures.
- ESM-c (ESM Team, 2024) scales up data and training compute, achieving performance improvements over ESM-2.
- ProtT5 (Elnaggar et al., 2020) trains T5 (Raffel et al., 2020), an auto-encoder language model, on UniRef and BFD data to capture biophysical features from protein sequences.
- ProtBert (Elnaggar et al., 2020) trains BERT (Devlin et al., 2019) on the UniRef and BFD data.
- S-PLM (Wang et al., 2024a) is a 3D structure-aware protein language model (PLM) that enables the sequence-based embedding to carry the structural information through multi-view contrastive learning.
- GPT-4o-mini (Hurst et al., 2024) is a large language model that demonstrates promising capabilities in understanding and generating human-like text.
- Pika (Carrami and Sharifzadeh, 2024) finetunes LLMs on a curated, debiased dataset tailored for protein question answering and a biochemically relevant benchmarking strategy. Protein embeddings by ESM-2 (Lin et al., 2022) are utilized to provide a comprehensive understanding of the target protein in the given question.

For similarity search algorithms, we first establish a reference database comprising enzymes from the training set. Test enzymes are then queried against this database, and the EC numbers of top-retrieved enzymes serve as predictions. For PLM baselines, we freeze the pre-trained weights while training a task-specific classification head based on the training set. The reported results for PLMs and PoinnCARE are averaged over five independent runs. We implement all baselines using their official code repositories as documented in Table 7, except for LLMs, which directly predict EC numbers through prompting.

## C.3 IMPLEMENTATION DETAILS

Accuracy is computed using the original code from CARE benchmark (Yang et al., 2024a), whereas precision, recall, and $F_1$ scores are calculated using the scikit-learn library (Pedregosa et al., 2011). Our PoinnCARE is implemented in Python using the PyTorch framework (Paszke, 2019). The model is optimized using the Adam optimizer (Kingma and Ba, 2014) with a learning rate of 0.01. The hidden dimension is set to 512, consistent with the baseline implementations. The hyperbolic encoders are implemented based on the Poincaré model with a fixed curvature of $\kappa = 1$. The graph diffusion operation over the active site similarity graph $G^{(a)}$ is instantiated as a two-layer personalized

Table 6: Summarization on the modality used in baselines.

| Category | Methods | Sequence | Modality Structure | Active site |
|---|---|:---:|:---:|:---:|
| Similarity search | BLASTp | ✓ | | |
| | Foldseek | | ✓ | |
| | Folddisco | | | ✓ |
| Contrastive learning | CLEAN | ✓ | | |
| | CLEAN-Concat | ✓ | ✓ | |
| PLM | ESM-2 | ✓ | | |
| | ESM-c | ✓ | | |
| | ProtT5 | ✓ | | |
| | ProtBert | ✓ | | |
| | S-PLM | ✓ | ✓ | |
| LLM | Pika | ✓ | | |

Table 7: The links to the baseline repositories.

| Category | Method | Github link |
|---|---|---|
| Similarity search | BLASTp | https://github.com/bbuchfink/diamond |
| | Foldseek | https://github.com/steineggerlab/foldseek |
| | Folddisco | https://github.com/steineggerlab/folddisco |
| Contrastive learning | CLEAN | https://github.com/tttianhao/CLEAN/tree/main |
| | CLEAN-Concat | https://github.com/PNNL-Predictive-Phenomics/clean-contact |
| PLM | ESM-2 | https://github.com/facebookresearch/esm |
| | ESM-c | https://github.com/evolutionaryscale/esm/tree/main |
| | ProtT5 | https://github.com/agemagician/ProtTrans |
| | ProtBert | https://github.com/agemagician/ProtTrans |
| | S-PLM | https://github.com/duolinwang/S-PLM/tree/main |

Table 8: Hyperparameters details.

|  | $\beta_s$ | $\beta_a$ | $\gamma$ | $W_d$ |
|---|---|---|---|---|
| <30% Identity test set | 0.8 | 0.2 | 0.001 | 0 |
| 30-50% Identity test set | 0.8 | 0.2 | 0.001 | 0 |
| Previously Misclassified (Price) test set | 0.2 | 0.8 | 0.0001 | 1 |
| Promiscuous test set | 0.5 | 0.5 | 0.0001 | 0 |

Table 9: Performance in terms of precision score on <30% Identity and 30-50% Identity test sets. The best and second-best results are shown in bold and underlined, respectively.

|  | <30% Identity | | | | 30-50% Identity | | | | Avg. rank |
|---|---|---|---|---|---|---|---|---|---|
|  | Level 1 (x.-.-.-) | Level 2 (x.x.-.-) | Level 3 (x.x.x.-) | Level 4 (x.x.x.x) | Level 1 (x.-.-.-) | Level 2 (x.x.-.-) | Level 3 (x.x.x.-) | Level 4 (x.x.x.x) | |
| Random* | 0.176 | 0.038 | 0.005 | 0.000 | 0.146 | 0.018 | 0.002 | 0.000 | 13.25 |
| BLASTp | 0.706 | 0.629 | 0.582 | 0.503 | 0.915 | 0.897 | 0.854 | 0.789 | 5.38 |
| Foldseek | 0.794 | 0.665 | 0.642 | 0.524 | 0.924 | 0.892 | 0.832 | 0.743 | 4.50 |
| Folddisco | 0.662 | 0.418 | 0.455 | 0.368 | 0.662 | 0.718 | 0.694 | 0.505 | 11.25 |
| CLEAN | 0.793 | 0.716 | 0.680 | 0.517 | 0.931 | 0.911 | 0.861 | 0.804 | 2.75 |
| CLEAN-Concat | 0.810 | 0.720 | 0.637 | 0.490 | 0.937 | 0.904 | 0.851 | 0.784 | 3.38 |
| ESM-2 | 0.785 | 0.675 | 0.620 | 0.486 | 0.943 | 0.883 | 0.819 | 0.770 | 4.88 |
| ESM-c | 0.690 | 0.548 | 0.527 | 0.402 | 0.902 | 0.854 | 0.778 | 0.738 | 9.00 |
| ProtT5 | 0.755 | 0.661 | 0.597 | 0.458 | 0.929 | 0.874 | 0.799 | 0.751 | 6.50 |
| ProtBert | 0.676 | 0.521 | 0.503 | 0.378 | 0.872 | 0.806 | 0.728 | 0.691 | 10.00 |
| S-PLM | 0.738 | 0.604 | 0.563 | 0.428 | 0.907 | 0.875 | 0.799 | 0.742 | 7.63 |
| ChatGPT* | 0.091 | 0.001 | 0.000 | 0.000 | 0.123 | 0.022 | 0.011 | 0.000 | 13.50 |
| Pika* | 0.587 | 0.483 | 0.357 | 0.196 | 0.714 | 0.588 | 0.498 | 0.354 | 11.75 |
| PoinnCARE | **0.885** | **0.826** | **0.778** | **0.616** | **0.951** | **0.924** | **0.875** | **0.818** | 1.00 |

PageRank algorithm with $\alpha = 0.8$. The values of other hyperparameters on four test sets are listed in Table 8, including:

- $\beta_s$ and $\beta_a$: trade-off parameters of modality fusion: $\boldsymbol{H} = \beta \cdot \boldsymbol{H}_{(s)} + \beta_a \cdot \boldsymbol{H}_{(a)}$.
- $\gamma$: trade-off parameters in the compound loss: $\mathcal{L} = \mathcal{L}_{clf} + \gamma \mathcal{L}_{align}$.
- $w_d$: weight of de-correlation terms in alignment loss 6.

# D  ADDITIONAL EXPERIMENTAL RESULTS

## D.1  PERFORMANCE REGARDING PRECISION, RECALL, AND $F_1$ SCORES

In this section, we provide additionally comprehensive evaluation results on other standard metrics: precision scores in Tables 9 and 10, recall scores in Tables 11 and 12, and $F_1$ scores in Tables 13 and 14.

The performance distributions across these metrics align with our previous accuracy analysis: our PoinnCARE demonstrates superior performance under most cases and maintains the highest average rank across all test sets, further validating the effectiveness of our hyperbolic multi-modal learning framework. Notably, on the Promiscuous set, BLASTp achieves marginally higher precision than our method, which can be attributed to the high sequence similarity between this test set and the training data (with nearly 50% of enzymes sharing >90% sequence identity). Nevertheless, PoinnCARE still surpasses BLASTp in both recall and $F_1$ metrics, achieving a substantial 4.4% improvement in level 4 $F_1$ score on the Promiscuous set.

## D.2  PERFORMANCE STABILITY

In this section, we provide the standard deviation results for the top three end-to-end prediction methods, PoinnCARE, ESM-2, and ProtT5, on the <30% Identity test set regarding accuracy. For the similarity search and contrastive learning baselines, the standard deviation is not applicable because

Table 10: Performance in terms of precision score on Price and Promiscuous test sets. The best and second-best results are shown in bold and underlined, respectively.

| | Previously Misclassidied (Price) | | | | Promiscuous | | | | Avg. rank |
|---|---|---|---|---|---|---|---|---|---|
| | Level 1 (x.-.-.-) | Level 2 (x.x.-.-) | Level 3 (x.x.x.-) | Level 4 (x.x.x.x) | Level 1 (x.-.-.-) | Level 2 (x.x.-.-) | Level 3 (x.x.x.-) | Level 4 (x.x.x.x) | |
| Random* | 0.220 | 0.051 | 0.007 | 0.000 | 0.363 | 0.081 | 0.036 | 0.005 | 13.13 |
| BLASTp | 0.818 | 0.784 | 0.696 | 0.341 | **0.968** | **0.960** | **0.951** | **0.874** | 4.25 |
| Foldseek | 0.939 | 0.878 | 0.797 | 0.318 | 0.846 | 0.808 | 0.799 | 0.704 | 4.25 |
| Folddisco | 0.000 | 0.000 | 0.000 | 0.000 | 0.813 | 0.740 | 0.693 | 0.458 | 12.13 |
| CLEAN | 0.858 | 0.794 | 0.686 | 0.307 | 0.909 | 0.873 | 0.848 | 0.691 | 5.38 |
| CLEAN-Concat | 0.905 | 0.868 | 0.767 | 0.348 | 0.917 | 0.860 | 0.834 | 0.659 | 3.75 |
| ESM-2 | 0.916 | 0.847 | 0.760 | 0.362 | 0.861 | 0.796 | 0.769 | 0.629 | 4.63 |
| ESM-c | 0.787 | 0.720 | 0.661 | 0.343 | 0.822 | 0.737 | 0.712 | 0.579 | 9.13 |
| ProtT5 | 0.891 | 0.824 | 0.757 | **0.380** | 0.846 | 0.766 | 0.734 | 0.596 | 6.38 |
| ProtBert | 0.674 | 0.551 | 0.512 | 0.200 | 0.794 | 0.691 | 0.658 | 0.549 | 11.00 |
| S-PLM | 0.867 | 0.788 | 0.718 | 0.238 | 0.851 | 0.780 | 0.756 | 0.606 | 7.13 |
| ChatGPT* | 0.372 | 0.176 | 0.095 | 0.000 | 0.292 | 0.086 | 0.067 | 0.005 | 12.63 |
| Pika* | 0.824 | 0.649 | 0.507 | 0.041 | 0.890 | 0.794 | 0.742 | 0.354 | 9.00 |
| PoinnCARE | **0.953** | **0.906** | **0.824** | 0.349 | 0.943 | 0.925 | 0.917 | 0.785 | 1.75 |

Table 11: Performance in terms of recall score on <30% Identity and 30-50% Identity test sets. The best and second-best results are shown in bold and underlined, respectively.

| | <30% Identity | | | | 30-50% Identity | | | | Avg. rank |
|---|---|---|---|---|---|---|---|---|---|
| | Level 1 (x.-.-.-) | Level 2 (x.x.-.-) | Level 3 (x.x.x.-) | Level 4 (x.x.x.x) | Level 1 (x.-.-.-) | Level 2 (x.x.-.-) | Level 3 (x.x.x.-) | Level 4 (x.x.x.x) | |
| Random* | 0.151 | 0.024 | 0.014 | 0.000 | 0.152 | 0.022 | 0.007 | 0.000 | 13.13 |
| BLASTp | 0.660 | 0.545 | 0.546 | 0.478 | 0.931 | 0.877 | 0.841 | 0.779 | 5.88 |
| Foldseek | 0.783 | 0.660 | 0.634 | 0.533 | 0.939 | 0.875 | 0.823 | 0.740 | 3.88 |
| Folddisco | 0.611 | 0.466 | 0.473 | 0.346 | 0.648 | 0.670 | 0.676 | 0.497 | 11.13 |
| CLEAN | 0.782 | 0.653 | 0.659 | 0.523 | 0.955 | 0.885 | 0.854 | 0.797 | 2.50 |
| CLEAN-Concat | 0.768 | 0.658 | 0.610 | 0.481 | 0.954 | 0.868 | 0.840 | 0.773 | 4.25 |
| ESM-2 | 0.745 | 0.632 | 0.618 | 0.494 | 0.955 | 0.868 | 0.828 | 0.771 | 4.63 |
| ESM-c | 0.650 | 0.507 | 0.505 | 0.413 | 0.925 | 0.839 | 0.782 | 0.737 | 9.00 |
| ProtT5 | 0.723 | 0.609 | 0.581 | 0.468 | 0.939 | 0.857 | 0.805 | 0.752 | 6.38 |
| ProtBert | 0.636 | 0.488 | 0.480 | 0.383 | 0.895 | 0.788 | 0.729 | 0.693 | 10.00 |
| S-PLM | 0.708 | 0.573 | 0.557 | 0.443 | 0.933 | 0.847 | 0.801 | 0.738 | 7.50 |
| ChatGPT* | 0.135 | 0.008 | 0.000 | 0.000 | 0.148 | 0.018 | 0.013 | 0.000 | 13.63 |
| Pika* | 0.581 | 0.402 | 0.355 | 0.187 | 0.763 | 0.562 | 0.477 | 0.350 | 11.88 |
| PoinnCARE | **0.885** | **0.787** | **0.761** | **0.632** | **0.969** | **0.909** | **0.870** | **0.825** | 1.00 |

Table 12: Performance in terms of recall score on Price and Promiscuous test sets. The best and second-best results are shown in bold and underlined, respectively.

| | Previously Misclassidied (Price) | | | | Promiscuous | | | | Avg. rank |
|---|---|---|---|---|---|---|---|---|---|
| | Level 1 (x.-.-.-) | Level 2 (x.x.-.-) | Level 3 (x.x.x.-) | Level 4 (x.x.x.x) | Level 1 (x.-.-.-) | Level 2 (x.x.-.-) | Level 3 (x.x.x.-) | Level 4 (x.x.x.x) | |
| Random* | 0.223 | 0.054 | 0.007 | 0.000 | 0.405 | 0.085 | 0.037 | 0.005 | 12.88 |
| BLASTp | 0.818 | 0.784 | 0.696 | 0.341 | 0.834 | 0.780 | 0.730 | 0.682 | 6.00 |
| Foldseek | 0.939 | 0.878 | 0.797 | 0.314 | 0.762 | 0.675 | 0.632 | 0.561 | 6.38 |
| Folddisco | 0.000 | 0.000 | 0.000 | 0.000 | 0.641 | 0.510 | 0.464 | 0.318 | 12.25 |
| CLEAN | 0.858 | 0.797 | 0.689 | 0.307 | 0.868 | 0.810 | 0.763 | 0.691 | 4.88 |
| CLEAN-Concat | 0.905 | 0.872 | 0.770 | 0.348 | 0.868 | 0.805 | 0.770 | 0.659 | 3.13 |
| ESM-2 | 0.918 | 0.849 | 0.762 | 0.362 | 0.857 | 0.771 | 0.716 | 0.629 | 4.00 |
| ESM-c | 0.791 | 0.726 | 0.666 | 0.343 | 0.813 | 0.722 | 0.665 | 0.579 | 8.13 |
| ProtT5 | 0.895 | 0.827 | 0.761 | **0.380** | 0.837 | 0.745 | 0.686 | 0.596 | 5.38 |
| ProtBert | 0.678 | 0.554 | 0.515 | 0.200 | 0.810 | 0.699 | 0.630 | 0.549 | 10.00 |
| S-PLM | 0.872 | 0.789 | 0.719 | 0.238 | 0.844 | 0.754 | 0.702 | 0.606 | 6.38 |
| ChatGPT* | 0.372 | 0.176 | 0.095 | 0.000 | 0.192 | 0.052 | 0.033 | 0.002 | 13.00 |
| Pika* | 0.824 | 0.649 | 0.507 | 0.041 | 0.611 | 0.465 | 0.362 | 0.164 | 11.00 |
| PoinnCARE | **0.955** | **0.909** | **0.827** | 0.349 | **0.908** | **0.866** | **0.844** | **0.785** | 1.25 |

Table 13: Performance in terms of $F_1$ score on <30% Identity and 30-50% Identity test sets. The best and second-best results are shown in bold and underlined, respectively.

| | <30% Identity | | | | 30-50% Identity | | | | Avg. rank |
|---|---|---|---|---|---|---|---|---|---|
| | Level 1 (x.-.-.-) | Level 2 (x.x.-.-) | Level 3 (x.x.x.-) | Level 4 (x.x.x.x) | Level 1 (x.-.-.-) | Level 2 (x.x.-.-) | Level 3 (x.x.x.-) | Level 4 (x.x.x.x) | |
| Random* | 0.150 | 0.023 | 0.007 | 0.000 | 0.143 | 0.018 | 0.003 | 0.000 | 13.13 |
| BLASTp | 0.673 | 0.551 | 0.528 | 0.484 | 0.921 | 0.872 | 0.830 | 0.778 | 5.63 |
| Foldseek | 0.785 | 0.638 | 0.610 | 0.520 | 0.930 | 0.871 | 0.810 | 0.736 | 4.25 |
| Folddisco | 0.604 | 0.430 | 0.448 | 0.354 | 0.648 | 0.673 | 0.665 | 0.498 | 11.13 |
| CLEAN | 0.783 | 0.661 | 0.641 | 0.511 | 0.942 | 0.885 | 0.842 | 0.795 | 2.63 |
| CLEAN-Concat | 0.782 | 0.662 | 0.592 | 0.480 | 0.944 | 0.873 | 0.830 | 0.772 | 3.63 |
| ESM-2 | 0.752 | 0.629 | 0.589 | 0.483 | 0.948 | 0.865 | 0.809 | 0.766 | 4.88 |
| ESM-c | 0.659 | 0.500 | 0.479 | 0.398 | 0.912 | 0.834 | 0.762 | 0.732 | 9.00 |
| ProtT5 | 0.730 | 0.604 | 0.555 | 0.455 | 0.932 | 0.854 | 0.785 | 0.745 | 6.25 |
| ProtBert | 0.642 | 0.478 | 0.456 | 0.373 | 0.880 | 0.779 | 0.703 | 0.685 | 10.00 |
| S-PLM | 0.714 | 0.559 | 0.525 | 0.427 | 0.917 | 0.846 | 0.783 | 0.735 | 7.75 |
| ChatGPT* | 0.085 | 0.002 | 0.000 | 0.000 | 0.100 | 0.013 | 0.008 | 0.000 | 13.63 |
| Pika* | 0.570 | 0.382 | 0.318 | 0.184 | 0.726 | 0.540 | 0.447 | 0.340 | 11.88 |
| PoinnCARE | **0.883** | **0.787** | **0.742** | **0.617** | **0.959** | **0.910** | **0.859** | **0.816** | 1.00 |

Table 14: Performance in terms of $F_1$ score on Price and Promiscuous test sets. The best and second-best results are shown in bold and underlined, respectively.

| | Previously Misclassidied (Price) | | | | Promiscuous | | | | Avg. rank |
|---|---|---|---|---|---|---|---|---|---|
| | Level 1 (x.-.-.-) | Level 2 (x.x.-.-) | Level 3 (x.x.x.-) | Level 4 (x.x.x.x) | Level 1 (x.-.-.-) | Level 2 (x.x.-.-) | Level 3 (x.x.x.-) | Level 4 (x.x.x.x) | |
| Random* | 0.221 | 0.052 | 0.007 | 0.000 | 0.364 | 0.080 | 0.036 | 0.005 | 13.00 |
| BLASTp | 0.818 | 0.784 | 0.696 | 0.341 | 0.879 | 0.840 | 0.804 | 0.746 | 4.75 |
| Foldseek | 0.939 | 0.878 | 0.797 | 0.315 | 0.783 | 0.717 | 0.686 | 0.609 | 5.50 |
| Folddisco | 0.000 | 0.000 | 0.000 | 0.000 | 0.688 | 0.582 | 0.541 | 0.366 | 12.38 |
| CLEAN | 0.858 | 0.795 | 0.687 | 0.307 | 0.871 | 0.823 | 0.789 | 0.691 | 5.25 |
| CLEAN-Concat | 0.905 | 0.869 | 0.768 | 0.348 | 0.877 | 0.817 | 0.789 | 0.659 | 3.63 |
| ESM-2 | 0.917 | 0.847 | 0.761 | 0.362 | 0.839 | 0.766 | 0.730 | 0.629 | 4.13 |
| ESM-c | 0.788 | 0.722 | 0.663 | 0.343 | 0.798 | 0.713 | 0.676 | 0.579 | 8.50 |
| ProtT5 | 0.892 | 0.825 | 0.759 | **0.380** | 0.820 | 0.736 | 0.698 | 0.596 | 5.63 |
| ProtBert | 0.676 | 0.552 | 0.513 | 0.200 | 0.778 | 0.679 | 0.634 | 0.549 | 10.25 |
| S-PLM | 0.869 | 0.788 | 0.719 | 0.238 | 0.826 | 0.749 | 0.716 | 0.606 | 6.63 |
| ChatGPT* | 0.372 | 0.176 | 0.095 | 0.000 | 0.224 | 0.063 | 0.044 | 0.002 | 12.88 |
| Pika* | 0.824 | 0.649 | 0.507 | 0.041 | 0.703 | 0.573 | 0.486 | 0.223 | 10.88 |
| PoinnCARE | **0.954** | **0.907** | **0.825** | 0.349 | **0.912** | **0.880** | **0.867** | **0.785** | 1.25 |

Table 15: The standard deviation values of the top three ene-to-end prediction methods.

|  | PoinnCARE | | ESM2 | | ProtT5 | |
|---|---|---|---|---|---|---|
|  | Mean | Std | Mean | Std | Mean | Std |
| x.-.-.- | 0.900 | 0.003 | 0.783 | 0.014 | 0.755 | 0.009 |
| x.x.-.- | 0.827 | 0.006 | 0.695 | 0.014 | 0.649 | 0.015 |
| x.x.x.- | 0.779 | 0.010 | 0.643 | 0.013 | 0.604 | 0.012 |
| x.x.x.x. | 0.648 | 0.006 | 0.518 | 0.015 | 0.492 | 0.018 |

Table 16: Performance under varying dimensions.

| Test set Dimension |  | <30% Identity | | | | | 30-50% Identity | | | | |
|---|---|---|---|---|---|---|---|---|---|---|---|
|  |  | 512 | 256 | 128 | 64 | 32 | 512 | 256 | 128 | 64 | 32 |
| CLEAN | x.-.-.- | 0.806 | 0.831 | 0.817 | 0.752 | 0.694 | 0.946 | 0.948 | 0.948 | 0.941 | 0.902 |
|  | x.x.-.- | 0.729 | 0.738 | 0.731 | 0.664 | 0.560 | 0.905 | 0.911 | 0.909 | 0.889 | 0.836 |
|  | x.x.x.- | 0.678 | 0.701 | 0.685 | 0.593 | 0.486 | 0.870 | 0.877 | 0.870 | 0.848 | 0.788 |
|  | x.x.x.x | 0.535 | 0.532 | 0.546 | 0.458 | 0.354 | 0.798 | 0.804 | 0.780 | 0.755 | 0.673 |
| PoinnCARE | x.-.-.- | 0.900 | 0.895 | 0.888 | 0.880 | 0.862 | 0.961 | 0.963 | 0.96 | 0.959 | 0.951 |
|  | x.x.-.- | 0.827 | 0.824 | 0.817 | 0.812 | 0.787 | 0.926 | 0.924 | 0.925 | 0.921 | 0.909 |
|  | x.x.x.- | 0.779 | 0.776 | 0.768 | 0.757 | 0.734 | 0.887 | 0.888 | 0.885 | 0.880 | 0.865 |
|  | x.x.x.x | 0.648 | 0.641 | 0.633 | 0.622 | 0.597 | 0.822 | 0.816 | 0.818 | 0.811 | 0.792 |

these methods are deterministic and do not exhibit variability across different runs. As shown in Table 15, our PoinnCARE demonstrates superior and also stable performance with low standard deviation values.

### D.3 PERFORMANCE UNDER VARYING DIMENSIONS

The accuracy performance of PoinnCARE and the strongest baseline, CLEAN, on the <30% and 30–50% test sets under varying dimensions, ranging from 512 down to 32, is presented in Table 16.

### D.4 PARAMETER ANALYSIS

**Inductive vs. transductive settings.** Our main experiments adhere to the inductive learning paradigm (Hamilton et al., 2017), enforcing strict information constraints whereby only training samples and their inter-relationships are accessible during the training phase. The learned model subsequently generalizes to previously unseen test instances during inference. We further investigate performance under the transductive setting (Kipf and Welling, 2016), wherein test samples are made available during training while their labels remain concealed. *The main distinction between these two paradigms lies in the exploitation of train-test similarity relationships in the training stage.* Table 17 presents accuracy comparisons across four test sets under both settings. Our PoinnCARE demonstrates robust generalizability even under the strict inductive learning paradigm.

Table 17: PoinnCARE performance under inductive and transductive settings regarding accuracy.

|  | <30% Identity | | | | 30-50% Identity | | | |
|---|---|---|---|---|---|---|---|---|
|  | x.-.-.- | x.x.-.- | x.x.x.- | x.x.x.x. | x.-.-.- | x.x.-.- | x.x.x.- | x.x.x.x. |
| Inductive | 0.900 | 0.827 | 0.779 | 0.648 | 0.961 | 0.926 | 0.887 | 0.822 |
| Transductive | 0.902 | 0.833 | 0.784 | 0.650 | 0.961 | 0.929 | 0.888 | 0.828 |
|  | Previously Misclassidied (Price) | | | | Promiscuous | | | |
|  | x.-.-.- | x.x.-.- | x.x.x.- | x.x.x.x. | x.-.-.- | x.x.-.- | x.x.x.- | x.x.x.x. |
| Inductive | 0.955 | 0.909 | 0.827 | 0.349 | 0.911 | 0.871 | 0.849 | 0.785 |
| Transductive | 0.962 | 0.917 | 0.829 | 0.359 | 0.915 | 0.876 | 0.852 | 0.790 |

Table 18: Performance under different hyperbolic space cuvatures.

| | Leanable $c$ | $c = -0.5$ | $c = -1$ | $c = -5$ |
|---|---|---|---|---|
| x.-.-.- | 0.896 | 0.894 | 0.902 | 0.900 |
| x.x.-.- | 0.826 | 0.832 | 0.833 | 0.825 |
| x.x.x.- | 0.781 | 0.780 | 0.784 | 0.777 |
| x.x.x.x. | 0.649 | 0.648 | 0.650 | 0.637 |

Table 19: Performance with variour graph diffusion settings.

| | | PPR | | | HKPR | | |
|---|---|---|---|---|---|---|---|
| | | $\alpha = 0.8$ | $\alpha = 0.5$ | $\alpha = 0.2$ | $t = 2$ | $t = 5$ | $t = 8$ |
| <30% Identity | x.-.-.- | 0.902 | 0.894 | 0.897 | 0.890 | 0.892 | 0.896 |
| | x.x.-.- | 0.833 | 0.834 | 0.837 | 0.832 | 0.833 | 0.836 |
| | x.x.x.- | 0.784 | 0.791 | 0.788 | 0.789 | 0.787 | 0.789 |
| | x.x.x.x. | 0.650 | 0.656 | 0.652 | 0.656 | 0.655 | 0.652 |
| 30-50 Identity | x.-.-.- | 0.961 | 0.966 | 0.964 | 0.967 | 0.965 | 0.965 |
| | x.x.-.- | 0.929 | 0.932 | 0.929 | 0.933 | 0.931 | 0.929 |
| | x.x.x.- | 0.888 | 0.891 | 0.889 | 0.890 | 0.889 | 0.889 |
| | x.x.x.x. | 0.828 | 0.832 | 0.831 | 0.830 | 0.830 | 0.830 |

**Curvature of the hyperbolic space.** In our main experiments, we fix the curvature of the underlying hyperbolic space to $c = -1$. The larger the absolute value of curvature, the more strongly curved the hyperbolic space becomes. The traditional Euclidean space corresponds to zero curvature ($c = 0$). We investigate performance sensitivity to different curvature values by setting $c$ to $-0.5$, $-5$, and a learnable parameter optimized jointly with the model. Table 18 presents accuracy results on the <30% Identity test set. Our findings indicate that curvatures closer to zero yield comparable performance, while $c = -5$ creates a strongly curved space that leads to slightly decreased accuracy.

**Graph diffusion.** To address the sparsity of enzyme active site annotations, we apply graph diffusion over the active site similarity graph $G^{(a)}$. Our main experiments employ a two-layer personalized PageRank (Wang et al., 2017) with $\alpha = 0.8$ as the graph diffusion mechanism. We further investigate alternative parameter configurations and diffusion instantiations, such as Heat Kernel PageRank (Kloster and Gleich, 2014), to evaluate the impact of different diffusion strategies. As presented in Table 19, PoinnCARE maintains superior performance across various graph diffusion settings, demonstrating the stability and effectiveness of our approach.

## E CASE STUDY

In this section, we present a case, D4APQ6, from the <30% Identity test set, to demonstrate how active site information helps complement structural information. D4APQ6 is from the <30% Identity test set, where its sequence identity with all training samples is deliberately restricted to less than 30%. The structural similarity with training samples returned by Foldseek also falls below the pre-defined threshold, preventing the formation of effective edges in the structural similarity graph. When relying solely on the structural graph, the enzyme D4APQ6 is *misclassified* as EC 3.2.1.67. However, Folddisco successfully identifies that D4APQ6 shares a similar local motif with the active sites (Cys-454 and Cys-457) of O22229 from the training set, as illustrated in Figure 10. This similarity results in the formation of homophilic

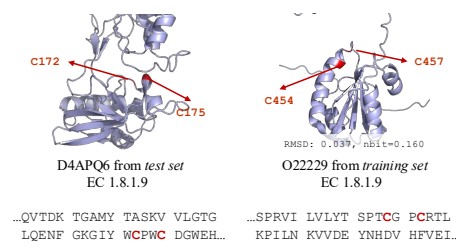

Figure 10: A correctly classified case.

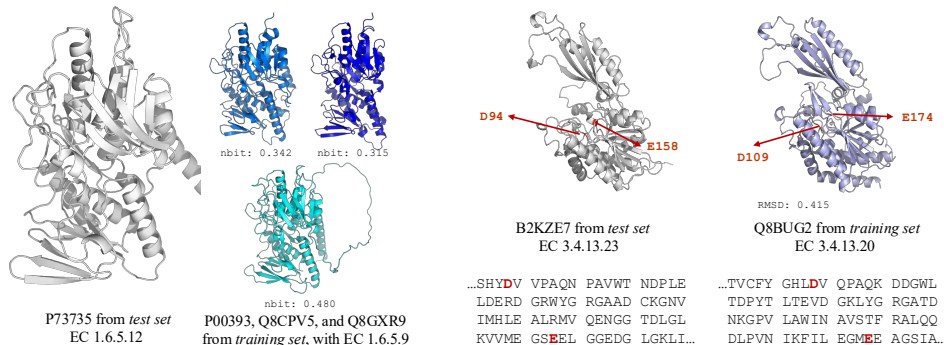

Figure 11: Examples of heterophilic structure (left) and active site (right) similarities, with the normalized bit score (nbit) and the motif RMSD provided.

edges in the active site similarity graph $G^{(a)}$, which facilitates information aggregation between test samples and known training samples, thereby enabling accurate classification of D4APQ6. Consequently, by integrating both structural and active site modalities, D4APQ6 is correctly classified as EC 1.8.1.9 by PoinnCARE.

## F  LIMITATION AND FUTURE WORK

In order to identify potential directions for future work, we first explore the reasons behind enzyme misclassification. Specifically, we collected enzymes from the <30% Identity test set that are misclassified by PoinnCARE. In this test set, sequence identity scores with training samples are strictly limited to below 30%. We further analyzed the structure and active site similarities associated with these misclassified enzymes and derived the following observations. First, we found that 53.3% (81 out of 152) of these misclassified enzymes lack both structural and active site similarity. Second, 45.3% (54 out of 152) of misclassified enzymes possess either structural similarity or active site similarity. However, the majority of these edges are heterophilic, meaning they connect to enzymes with different EC numbers. The homophily ratio among these edges is only 0.078, which is significantly lower than the overall homophily ratios. The absence of effective information and the presence of misleading, heterophilic relationships make the classification of these enzymes particularly challenging.

We further examined those enzymes that are misclassified at the deepest EC level, and present the examples where our model can be misled by heterophilic edges. Specifically, enzyme P73735, whose true EC number is 1.6.5.12, was misclassified as EC 1.6.5.9, as three out of four edges in the structural similarity graph connect to enzymes with EC 1.6.5.9 (Q8CPV5, Q8GXR9, and P00393), as shown in the left part of Fig. 11. Similarly, enzyme B2KZE7, with true EC number 3.4.13.23, was incorrectly assigned EC 3.4.13.20. In this case, the only edge for B2KZE7 in the active site similarity graph links to Q8BUG2 from the training set, which also has an EC number of 3.4.13.20. The right part of Fig. 11 demonstrates the identified similar local motifs. Based on the above observations, we believe that improving both the quantity and quality of similarity relationships—especially through more precise structural and active site information—will enhance performance in these difficult cases.

## G  BROADER IMPACTS

Our method contributes to more effective enzyme function prediction, which could facilitate the understanding of enzyme roles in various biological processes. The improved prediction accuracy has potential implications for both fundamental research in biotechnology and downstream industrial applications.

## H LLM USAGE

Per author guidelines, we disclose the use of Large Language Models for only writing polish.

