# OpenReview forum: "PoinnCARE: Hyperbolic Multi-Modal Learning for Enzyme Classification"
_ICLR.cc/2026/Conference — ICLR 2026 Poster_

### Official Review · Reviewer_B891 · 2025-10-24

**Soundness:** 3
**Presentation:** 3
**Contribution:** 3
**Rating:** 4
**Confidence:** 3

**Summary:**

This paper introduces PoinnCARE, a hyperbolic multi-modal learning framework for enzyme function prediction. By combining information from enzyme sequences, structures, and active sites, the approach leverages graph diffusion and dual-graph alignment within hyperbolic space to encode hierarchical relations informed by the Enzyme Commission (EC) system. Extensive experiments on the CARE benchmark demonstrate that PoinnCARE achieves improved accuracy over a strong set of baselines across several challenging test splits.

**Strengths:**

1. Integration of multiple modalities: The paper goes beyond sequence-based modeling by integrating experimentally determined or predicted enzyme structures and active site annotations, which addresses the intrinsic complexity of enzyme function prediction.
2. Hyperbolic embedding with theoretical justification: Embedding enzyme representations in hyperbolic space is both theoretically motivated  and practically validated. The paper rigorously discusses why hyperbolic geometry is ideal for modeling the EC system's hierarchy.
3. Comprehensive and robust empirical evaluation: The method is benchmarked across four different and increasingly challenging test scenarios from CARE, with comparisons to a wide range of recent and classical baselines, including CLEAN, CLEAN-Concat, various PLMs, and LLM approaches. Tables 2, 3, and supplemental metrics (precision, recall, F1; Tables 9–14) consistently show improvements of PoinnCARE, validating its robustness to sequence diversity and test set complexity.
4. Ablation and dimensionality studies: The results in Figure 6 and Table 16 demonstrate that performance gains are attributable to each design component, and PoinnCARE's effectiveness persists even at smaller embedding dimensions, reinforcing the theoretical claims regarding hyperbolic geometry.
5. Reproducibility efforts: The authors provide detailed data and implementation information, facilitating future reproduction.

**Weaknesses:**

1. Lack of comparison with advanced models: This paper uses enzyme structure and active site information, but the specialized models it compares against are limited to the sequence-based CLEAN and its derivatives. It does not include comparisons with structure-based models such as EnzymeCAGE (Liu et al. 2024) and GraphEC (Song et al. 2024).
(1) Liu, Yong, et al. "EnzymeCAGE: a geometric foundation model for enzyme retrieval with evolutionary insights." bioRxiv (2024): 2024-12.
(2) Song, Yidong, et al. "Accurately predicting enzyme functions through geometric graph learning on ESMFold-predicted structures." Nature Communications 15.1 (2024): 8180.
2. Questionable choice of hyperbolic GNN architecture: PoinnCARE adopts the Poincaré Ball Model and performs GNN aggregation and transformation in the tangent space at the origin ($o$). The authors state this choice was made "For simplicity". However, HGCN (Chami et al., 2019) explicitly argues for and demonstrates the superiority of aggregation in the local tangent space of each center node ($x_i^H$). PoinnCARE's architectural choice seems to be a simplification that may lead to suboptimal representation power. The paper does not provide sufficient justification that this "simple" choice is reasonable or equivalent to local-space aggregation.
(1) Chami, Ines, et al. "Hyperbolic graph convolutional neural networks." Advances in neural information processing systems 32 (2019).
3. Concerns about the efficacy of the active site modality: The paper admits that the active site data is extremely sparse, with "more than half of the enzymes have no annotated active site residues". Despite using graph diffusion, the information foundation for this modality is very weak. Furthermore, the input features $H^{(0)}$ for both hyperbolic GNNs are identical. This introduces a risk: the representation $H_{(a)}$ learned by the GNN on $G^{(a)}$ ($f_{hyp}^{(a)}$) might just be a variant of the $H^{(0)}$ features on a highly smoothed (post-diffusion) graph structure, without capturing much genuinely unique "active site" functional signal. The contribution of this modality may be overestimated.
4. Oversimplified modality fusion: The final fusion method for prediction is a simple weighted sum: $\beta_{s} \cdot H_{(s)} + \beta_{a} \cdot H_{(a)}$. Given the heterogeneous nature and different sparsity levels of the two modalities (structure vs. active site), this linear, static fusion method is likely suboptimal. The paper does not explore more dynamic fusion mechanisms (e.g., attention, gating).
5. Positioning in relation to closest prior work: Several highly relevant recent papers are not cited or discussed, particularly those specifically focusing on hyperbolic multimodal taxonomy or hierarchical enzyme function prediction. This weakens the contextualization and differentiation of the method’s originality.
(1) Gong, ZeMing, et al. "Hyperbolic Multimodal Representation Learning for Biological Taxonomies." arXiv preprint arXiv:2508.16744 (2025).
(2) Li, Nan, et al. "Hyperbolic hierarchical knowledge graph embeddings for biological entities." Journal of Biomedical Informatics 147 (2023): 104503.
6. Insufficient analysis of failure cases: The analysis leans heavily on aggregate metrics. It would be helpful to present examples or qualitative analysis, where the method fails, especially at deeper EC levels. There’s a lack of introspection on possible errors, such as misclassification of enzymes with convergent functions but divergent structures or sequences (a case hinted at by Figure 3, but not explored in error analysis).

**Questions:**

1. Could the authors please clarify the reasoning behind the exclusion of state-of-the-art specialized models from the evaluation?
2. What is the rationale for choosing to perform GNN operations in the origin's tangent space rather than the local node's tangent space? Beyond "simplicity," is there any theoretical or experimental evidence (e.g., a comparison of these two aggregation methods within the PoinnCARE framework) to support this decision?
3. Given the extreme sparsity of $G^{(a)}$ and the shared $H^{(0)}$ input, how can the authors demonstrate that $H_{(a)}$ has learned "active site"-specific information that is meaningful for prediction and distinct from $H_{(s)}$? Is there any qualitative or quantitative analysis of the differences between the $H_{(a)}$ and $H_{(s)}$ embedding spaces (e.g., t-SNE visualization or mutual information analysis)?
4. Why was a simple linear weighted sum chosen as the final modality fusion method? Did the authors experiment with more complex fusion mechanisms (e.g., concat + MLP, or cross-modal attention)?
5. In constructing $G^{(s)}$ and $G^{(a)}$ (Sections B.1 and B.2), how were the similarity thresholds $\delta^s=0.3$ and $\delta^a=0.05$ chosen? How sensitive is the model's performance to these thresholds?
6. Could the authors analyze cases where PoinnCARE makes errors at the deepest EC digit level (level 4), and provide qualitative examples? Are there biological patterns (e.g., convergent evolution) that systematically challenge the model?
7. How do the computational and memory costs of the hyperbolic GNN approach compare empirically to standard Euclidean GNNs, given the noted $O(nd)$ overhead?

---

> ### Author Response · Authors · 2025-11-21
>
> We deeply appreciate the reviewer for the effort and the time in reviewing our work and proposing detailed and valuable feedback. Our point-to-point responses are as follows.
>
> ---
>
> **Response to Weakness 1 and Question 1 (RW1 & RQ1).** EnzymeCAGE employs a different problem formulation compared to our approach. Specifically, EnzymeCAGE frames enzyme function prediction as **a binary classification task**, where given an (enzyme, reaction) pair, the model determines whether the enzyme can catalyze the specific reaction. In contrast, our work follows the previous work and formulates enzyme function prediction as a **multi-label and multi-class EC number classification problem** [r1, r2]. This fundamental difference in problem formulation further extends to distinctions in dataset construction and organization, creating the potential for **data leakage in direct comparison**, as EnzymeCAGE's training samples may overlap with the enzymes present in our test sets, thereby compromising the evaluation fairness.
>
>
> For GraphEC, the Price dataset also serves as a test set in their original study, providing a common benchmark for direct and fair comparison with PoinnCARE. We present the comparative results on this shared test set in the following table, where our PoinnCARE clearly and consistently **outperforms GraphEC across all metrics.** As the official evaluation code for GraphEC was not released, we prioritized standard ranking-based metrics, Area Under the Curve (AUC) and average precision (AP). Specifically, the AUC score of the Price dataset reported **in the original GraphEC paper is 0.840, while our PoinnCARE achieves an AUC of 0.868**, resulting in a 3.33% improvement. Regarding AP, our PoinnCARE outperforms GraphEC by 45.60%. Additionally, we also provide accuracy across four levels, aligning with the metrics used in our manuscript, where PoinnCARE surpasses GraphEC across all four levels.
>
>
> Table r1: Comparison results on the Price test set (accuracy).
>
> |  | AUC | AP | Level 1 | Level 2 | Level 3 | Level 4 |
> |---|---|---|---|---|---|---|
> | PoinnCARE | 0.868 | 0.182 | 0.955 | 0.909 | 0.827 | 0.349 |
> | GraphEC | 0.840 | 0.125 | 0.696 | 0.662 | 0.622 | 0.196 |
> | Improvement |  +3.33% | 45.60% | +37.22% | +37.28% | +33.04% | +78.11% |
>
>
> [r1] Yu, Tianhao, et al. "Enzyme function prediction using contrastive learning." Science 379.6639 (2023): 1358-1363.
>
> [r2] Yang, Jason, et al. "Care: a benchmark suite for the classification and retrieval of enzymes." NIPS, 2024.
>
> ---
>
> **RW2 & RQ2.** First, the origin-based tangent aggregation method is **widely adopted in previous studies** including [r1, r2, r3, r4], demonstrating its simplicity and efficiency.
>
> Second, the time and space complexity of center node-aggregation is substantially higher than that of origin-based aggregation. Compared to origin-based aggregation, center node-based aggregation in HGCN **increases the time and space complexity by a factor of $n$**, where $n$ represents the number of enzymes. However, this operation cannot bring consistent and notable performance improvement, as acknowledged by the authors of HGCN when responding to an issue at their code repository [r5]. For instance, on the AIRPORT dataset, the AUC difference between origin-based and center node-based aggregation is only **0.2%**, as shown by the ablation study in HGCN [1].
>
> We optimized the HGCN code to reduce the complexity overhead from $n$ to $d_i$, where $n$ and $d_i$ denote the number of nodes and the degree of the center node $i$ in the similarity graphs. However, this operation still significantly extends the running time (from 0.16s per epoch to 148s per epoch). Thus, the experiments are still ongoing, and we will supplement the comparison results under these two configurations once the experiments are completed.
>
>
> [r1] Zhang, Yiding, et al. "Hyperbolic graph attention network." IEEE Transactions on Big Data 8.6 (2021): 1690-1701.
>
> [r2] Dong, Xiangyu, et al. "Spacegnn: Multi-space graph neural network for node anomaly detection with extremely limited labels." ICLR, 2025.
>
> [r3] Yang, Menglin, et al. "Hyperbolic representation learning: Revisiting and advancing." ICLR, 2023.
>
> [r4] Liu, Qi, Maximilian Nickel, and Douwe Kiela. "Hyperbolic graph neural networks." NIPS, 2019.
>
> [r5] https://github.com/HazyResearch/hgcn/issues/18

---

> ### Author Response · Authors · 2025-11-21
>
> **RW3 & RQ3.** To demonstrate the contribution of active site modality to enzyme EC number prediction, we provide additional top-down ablation results by separately removing the active site and structural modalities from PoinnCARE on the <30% Identity test set, as shown in the following table. As we can see, **removing the active site modality results in a 4.3% degradation** in level 4 accuracy, which demonstrates that active site information makes a substantial contribution to classification performance.
>
>
> Table r2: Ablation results after removing the active site or structural modality.
>
> |  | Level 1 | Level 2 | Level 3 | Level 4 |
> |---|---|---|---|---|
> | PoinnCARE | 0.902 | 0.833 | 0.784 | 0.650 |
> | PoinnCARE w/o active site modal | 0.875 (3.04% ↓) | 0.812 (2.57% ↓) | 0.765 (2.39% ↓) | 0.622 (4.34% ↓) |
> | PoinnCARE w/o structure modal | 0.865 (4.12% ↓) | 0.794 (4.63% ↓) | 0.745 (4.93% ↓) | 0.613 (5.70% ↓) |
>
>
> We further examined the samples that were misclassified using the single structural graph with sequence features but correctly classified using the dual graphs. Specifically, we present a representative case in Appendix E in the revised manuscript: D4APQ6 from the <30% Identity test set. Due to its restricted sequence identity and the absence of effective structural relationships, D4APQ6 is challenging to classify using either sequence features or a single structural graph.  However, Folddisco successfully identifies similarity between a local motif of D4APQ6 and known active sites from the training set O22229, as shown in Figure 10. This similarity establishes homophilic edges and facilitates information aggregation between test samples and known training samples. Consequently, by integrating active site modalities with structural similarity and sequence information, D4APQ6 is correctly classified by PoinnCARE.
>
> To summarize, this analysis reveals that structural and active site modalities can **capture complementary information**, and integrating both modalities allows the model to retain the strengths of each, leading to improved overall classification performance.
>
> Finally, we present a Venn diagram in Figure 9 of our revised manuscript, illustrating the distribution of structural and active site graphs. The **modest number of shared relationships** revealed by this diagram indicates that each modality captures distinct and complementary information.
>
> ---
>
> **RW4 & RQ4.** We evaluated two fusion methods for PoinnCARE, weighted sum and concatenation, on two test sets, with accuracy results detailed in the following Table. While the concatenation method resulted in slightly lower accuracy under several cases, it surpassed the weighted sum approach for level-4 accuracy on the 30-50 Identity test set by 1%, demonstrating noteworthy potential. Additionally, weighted sum fusion is also utilized in previous works, including [r1, r2, r3].
>
> Table r3: Performance with different fusion functions.
>
> |  |  | Level 1 | Level 2 | Level 3 | Level 4 |
> |---|---|---|---|---|---|
> | <30 Identity | weighted sum | 0.900 | 0.827 | 0.779 | 0.648 |
> |  | concatenation | 0.879 | 0.799 | 0.756 | 0.622 |
> | 30-50 Identity | weighted sum | 0.961 | 0.926 | 0.887 | 0.822 |
> |  | concatenation | 0.956 | 0.920 | 0.882 | 0.830 |
>
> [r1] Ren, Yazhou, et al. "Multi-View Graph Clustering via Node-Guided Contrastive Encoding."ICLR, 2025.
>
> [r2] Xie, Kun, Renchi Yang, and Sibo Wang. "Diffusion-based Graph-agnostic Clustering." WWW, 2025.
>
> [r3] Pan, Erlin, and Zhao Kang. "Beyond homophily: Reconstructing structure for graph-agnostic clustering." International conference on machine learning. ICML, 2023.
>
>
> ---
>
> **RW5.** We thank the reviewer for highlighting these recent studies. In response, we have updated our manuscript to **include a more comprehensive introduction to related works**, encompassing not only existing methods for enzyme function prediction but also advances in hyperbolic representation learning, with a particular emphasis on its applications within the biological domain.
>
> Specifically, Gong et al. [1] have integrated multiple modalities, including DNA barcodes, specimen images, and taxonomic labels, into a shared hyperbolic space to facilitate taxonomic alignment. Li et al. [2] have embedded the rich knowledge derived from Gene Ontology (GO) within hyperbolic space, yielding accurate representations of biological entities. While hyperbolic learning has been explored in various biological applications to enhance the analysis of genomes, entities, and biomolecular affinity, **the exploration involving enzymes and the EC number systems remains underdeveloped**, particularly when considering the integration of the rich information associated with each enzyme, such as sequence, structure, and active site data.

---

> ### Author Response · Authors · 2025-11-21
>
> **RW6 & RQ6.** Following the reviewer’s suggestion, we collected enzymes from the <30% Identity test set that are *misclassified by PoinnCARE*. In this test set, sequence identity scores with training samples are strictly limited to below 30%. We further analyzed the structure and active site similarities associated with these misclassified enzymes and derived the following observations.
>
> First, we found that 53.3% (81 out of 152) of these misclassified enzymes **lack both structural and active site similarity**. The absence of effective sequence, structural, and active site information makes the classification of these enzymes particularly challenging.
>
> Second, 35.5% (54 out of 152) of misclassified enzymes possess either structural similarity or active site similarity. However, the majority of these edges are **heterophilic**, meaning they connect to enzymes with different EC numbers. The homophily ratio among these edges is only 0.078, which is significantly lower than the overall homophily ratios of 0.808 and 0.514 for graphs $G^s$ and $G^a$, respectively. The presence of misleading, heterophilic structural or active site similarity, especially under restricted sequence identity, further complicates accurate EC prediction.
>
> We further examined those enzymes that are **misclassified at the deepest EC level.** These errors can also be primarily attributed to insufficient information (45.6%) or the misleading influence of heterophilic edges (43.9%). The following examples specifically demonstrate how heterophilic edges can mislead the model.
>
> - Enzyme P73735 (true EC: 1.6.5.12) was misclassified as EC 1.6.5.9. Upon analysis, we found that three out of four edges in the structural similarity graph connect to enzymes with EC 1.6.5.9, including Q8CPV5, Q8GXR9, and P00393. A structural comparison is presented in the left part of Figure 11 of our revised manuscript.
> - Enzyme B2KZE7 (true EC: 3.4.13.23) was incorrectly assigned EC 3.4.13.20. The only edge for B2KZE7 links to Q8BUG2 from the training set, which has EC 3.4.13.20. Local motif comparison is provided in the right part of Figure 11 of the revised manuscript.
>
> Based on the above observations, we believe that improving both the quantity and quality of similarity relationships, especially through more precise structural and active site information, will enhance performance in these difficult cases. This also aligns with the future directions proposed in our original manuscript, where we highlight the refinement of active site data in future work. The integration of additional key characteristics involved in catalysis, such as substrate and reaction details, would provide complementary information and further enhance prediction performance.
>
> Additionally, we provide a **positive example** in which enzyme D4APQ6 is initially misclassified when using only the structural similarity graph, but is correctly classified through the integration of both structural and active site similarity graphs. In this case, **homophilic edges** identified by Folddisco help compensate for limited sequence and structural information, as detailed in **RW3** and Appendix E of our revised manuscript.

---

> ### Author Response · Authors · 2025-11-21
>
> **RQ5.** The threshold values are chosen based on the homophily degree of the training similarity graphs. Specifically, for $\delta^s$,  which denotes the threshold of the normalized sequence identity, setting a higher value leads to fewer edges and increases the homophily ratio due to more stringent edge formation criteria. In contrast, a lower threshold results in more edges, decreasing the homophily ratio due to the inclusion of possible noisy edges arising from a more relaxed standard. Therefore, it is important to select a threshold that balances the quantity and quality of edges. The effects of varying $\delta^a$, the RMSD threshold of local motifs, follow similar trends.
>
> In our study, we selected $\delta^s=0.3$ and $\delta^a=0.05$ by **observing the changes in the number of edges and homophily ratios among the training samples** during preprocessing. Additionally, we evaluated performance across varying threshold values. Accuracy results for the <30% Identity test set are presented in the following two tables.
>
> For the threshold $\delta^s$ of the structural similarity graph, we tested values of 0.1, 0.3, 0.5, 0.7, and 0.9, with PoinnCARE selecting 0.3 as the threshold. PoinnCARE under a single structural modality was utilized for a clearer assessment of the impact of threshold values. As shown in the table, as the threshold increases, the performance first improves, reaches the peak at $\delta^s=0.5$, and subsequently declines.
>
> Table r4: PoinnCARE accuracy under a single structural modality with varying thresholds.
>
> | $\delta^s$ | Level 1 | Level 2 | Level 3 | Level 4 |
> |---|---|---|---|---|
> | 0.1 | 0.881 | 0.782 | 0.719 | 0.550 |
> | 0.3 | 0.875 | 0.812 | 0.765 | 0.622 |
> | 0.5 | 0.877 | 0.805 | 0.762 | 0.628 |
> | 0.7 | 0.875 | 0.801 | 0.751 | 0.619 |
> | 0.9 | 0.863 | 0.793 | 0.746 | 0.611 |
>
> For the threshold $\delta^a$ of the active site similarity graph, we tested the values of 0.01, 0.05, 0.1, and 0.5, with PoinnCARE selecting 0.05 as the threshold. The performance variance over different $\delta^a$ values is similar to that over $\delta^s$, with a performance peak observed at $\delta^a=0.1$.
>
> Table r5: PoinnCARE accuracy under a single active site modality with varying thresholds.
>
> | $\delta^a$ | Level 1 | Level 2 | Level 3 | Level 4 |
> |---|---|---|---|---|
> | 0.01 | 0.871 | 0.792 | 0.739 | 0.613 |
> | 0.05 | 0.865 | 0.794 | 0.745 | 0.613 |
> | 0.1 | 0.865 | 0.79 | 0.739 | 0.615 |
> | 0.5 | 0.857 | 0.775 | 0.725 | 0.603 |
> | 0.9 | 0.863 | 0.793 | 0.746 | 0.611 |
>
> ---
> **RQ7.** The hyperbolic operations do not introduce additional parameters to the standard GNN, leading to a comparative space cost, which can be handled by a single GPU with 40GB memory. For the running time, under identical hardware conditions and dimensional settings, the standard GCN on our dual similarity graphs requires 1,743.7s for training and testing, whereas our PoinnCARE requires 2,154.6s.
>
> ---
>
> We hope these responses address your concerns, and we are more than happy to answer any questions you may have.

---

> ### Comment · Reviewer_B891 · 2025-11-26
>
> Thank you for the detailed response. I have a follow-up question regarding the ablation study presented in Figure 7. What embedding dimension was used for this specific experiment? Could the authors investigate the performance changes if the hidden layer dimension is increased in the Euclidean setting (i.e., without introducing the hyperbolic module)? While I acknowledge the theoretical advantage of hyperbolic space in achieving low distortion for low-dimensional embeddings, modern GPUs can effortlessly handle Euclidean vectors of 512 dimensions or even higher. Therefore, it is important to verify if the hyperbolic complexity is truly necessary when high-dimensional Euclidean space is available. Additionally, I noted that the paper does not specify the computing platform or hardware resources used.

---

> ### Author Response · Authors · 2025-11-26
>
> We appreciate your time in reviewing our responses and revisions.
>
> Regarding the ablation study presented in Figure 7, we maintained an embedding dimension of 512 for all incomplete variants used in this experiment.
>
> To determine if increasing Euclidean dimensions can compensate for embedding distortion and enhance performance, we systematically increased the dimensionality of the Euclidean space while keeping all other components of our PoinnCARE model unchanged. This Euclidean variant, referred to as *Euc*, was tested across dimensions of 512, 1024, and 2048, with accuracy evaluated on the <30% Identity test set. For comparison, results from the Hyperbolic setting with a default dimension of 512, denoted as *Hyp*, are also provided in the following table.
>
> Increasing the dimensions of the Euclidean variant from 512 to 2048 resulted in an accuracy improvement of 3.22% (from 0.621 to 0.641) at level 4. However, this **came with an increase in running time of 75.57%** (from 1904.8 s to 3344.3 s). Notably, the Euclidean variant at dimension 2048 achieved a level 4 accuracy of 0.641, while the Hyperbolic variant of dimension 512 achieved a higher accuracy of 0.650 (1.4% ↑ ) with a reduced running time of 2332.9 s (30.24% ↓). These results particularly highlight the superior effectiveness and efficiency of utilizing Hyperbolic space. In summary, **increased dimensionality in Euclidean space improves performance at the cost of significantly increased time, ultimately failing to beat the performance in hyperbolic space.**
>
>
> Table r6: Performance under the Euclidean setting with increasing dimensions.
>
> | dim | 512 | 1024 | 2048 | 512 |
> |---|:---:|:---:|:---:|:---:|
> | variant | Euc | Euc | Euc | Hyp |
> | Level 1 | 0.871 | 0.882 | 0.881 | 0.902 |
> | Level 2 | 0.797 | 0.811 | 0.814 | 0.833 |
> | Level 3 | 0.745 | 0.752 | 0.759 | 0.784 |
> | Level 4 | 0.621 | 0.628 | 0.641 | 0.650 |
> | Time(s) | 1904.8 | 2563.1 | 3344.2 | 2332.9 |
>
> Note: *Hyp* indicates the PoinnCARE model itself, while *Euc* represents its variant configured in Euclidean space.

---

> > ### Author Response · Authors · 2025-11-26
> >
> > Regarding the hardware setup, all experiments were conducted on a single GPU with 40GB of memory, offering peak performance of 19.5 TFLOPS in FP32, which is suitable for AI research. We are unable to disclose the specific model due to certain reasons. We sincerely appreciate your understanding.
> >
> > We hope these responses address your concerns, and we are more than happy to answer any more questions you may have.

---

> ### Comment · Reviewer_B891 · 2025-11-27
>
> Thanks for your responses. My concerns are addressed and I have adjusted the scores.

---

> > ### Author Response · Authors · 2025-11-27
> >
> > We sincerely appreciate you taking the time to review our paper and offer your valuable support.

---

### Official Review · Reviewer_e3qB · 2025-10-31

**Soundness:** 3
**Presentation:** 3
**Contribution:** 4
**Rating:** 8
**Confidence:** 3

**Summary:**

This paper presents PoinnCARE, a novel framework for Enzyme Commission (EC) number prediction. It compellingly argues that existing methods fail by (1) not adequately modeling the hierarchical, tree-like structure of the EC classification system and (2) overlooking critical multi-modal data, specifically protein structure and active site information. The core proposal is to solve this by embedding multi-modal enzyme data (sequence, structure, and active site) into hyperbolic space, which is theoretically well-suited for hierarchical data.

**Strengths:**

(1) The paper's greatest strength is its clear and theoretically-grounded motivation. The introduction clearly identifies two major limitations of prior work: ignoring the EC hierarchy and overlooking multi-modal data . They also provide a strong theoretical case for why using hyperbolic space.

(2) The experimental section is thorough, rigorous, and provides powerful evidence for the paper's claims. PoinnCARE consistently achieves the highest accuracy in nearly all cases. The authors have also provided extensive ablation studies to support the effectivness of their framework.

**Weaknesses:**

The most significant con I see of this paper is that it seems to require heavy hyperparmeters tuning. Table 8 lists the final hyperparameters, which is good for reproducibility. However, it reveals a potential weakness: the modality-weighting parameters are different for each test set.

Before any learning occurs, the framework requires running multiple computationally intensive bioinformatics tools. This includes using Foldseek to compute all-vs-all structural similarity and Folddisco to compute all-vs-all active site motif similarity. This is more complex than a model that simply takes a protein sequence as input.

**Questions:**

Is it possible to further simplify the framework so that the overall complexity could be lower.

---

> ### Author Response · Authors · 2025-11-21
>
> We sincerely thank the reviewer for the time and effort in evaluating our work. We greatly appreciate the insightful comments and helpful suggestions. Our point-to-point responses are as follows.
>
> ---
>
> **RW1.** Thank you for bringing up this concern. Our preliminary empirical results suggest that suitable values for the modality-weighting parameters $\beta_s$ and $\beta_a$ can be selected from the following three combinations: (0.8, 0.2), (0.5, 0.5), and (0.2, 0.8). We observed that these options can effectively cover most practical cases in our experiments.
>
> ---
>
> **RW2.** Thank you for pointing out the additional complexity involved in the preprocessing stage of our framework. While our framework entails additional preprocessing with Foldseek and Folddisco, the construction of the training databases in Foldseek and Folddisco is a one-time cost that can be reused for any subsequent query. Furthermore, both Foldseek and Folddisco are highly optimized for speed, allowing users to process large datasets within a reasonable time frame. To further facilitate adoption, we will provide a unified pipeline script so that all steps, from preprocessing to inference, can be executed seamlessly.
>
> ---
>
> **RQ1.** Thank you for your insightful question. The overall complexity of our pipeline primarily originates from the additional preprocessing steps and the incorporation of hyperbolic operations. A potential variant that learns to automatically mine similar structures and active sites directly from sequence inputs would substantially enhance usability for new users by reducing the need for manual preprocessing. Furthermore, approaches such as sampling, distillation, and quantization are promising for improving the scalability of the hyperbolic GNN component; these strategies have already been validated in previous works for standard GNN frameworks [r1, r2, r3]. We consider exploration of these directions as promising future work to further optimize and simplify our method.
>
>
> [r1] Chen, Jie, Tengfei Ma, and Cao Xiao. "Fastgcn: fast learning with graph convolutional networks via importance sampling." ICLR, 2018.
>
> [r2] Guo, Zhichun, et al. "Linkless link prediction via relational distillation." International conference on machine learning. ICML, 2023.
>
> [3] Ding, Mucong, et al. "Vq-gnn: A universal framework to scale up graph neural networks using vector quantization." NIPS. 2021.
>
> ---
>
> We thank the reviewer again for the valuable feedback. We are glad to answer any additional questions you may have.

---

> > ### Comment · Reviewer_e3qB · 2025-11-25
> >
> > Thank you for the response. While I remain slightly skeptical about the hyperparameter tuning part I don't think it harms the paper much either. I'll remain my score.

---

> > > ### Author Response · Authors · 2025-11-26
> > >
> > > We truly appreciate you taking the time to review our paper and offer your valuable support.

---

### Official Review · Reviewer_znFn · 2025-11-02

**Soundness:** 3
**Presentation:** 3
**Contribution:** 3
**Rating:** 8
**Confidence:** 4

**Summary:**

This paper introduces PoinnCARE, a hyperbolic multi-modal learning framework for enzyme function classification. PoinnCARE integrates three modalities — protein sequence, 3D structure, and active site motif — into a unified representation. The representations are used to train a hyperbolic GNN in order to model the hierarchical relationships of the Enzyme Commission (EC) classification tree. Trained on an updated CARE benchmark dataset, PoinnCARE achieves state-of-the-art enzyme classification accuracy across several test sets, including low-homology and promiscuous enzymes.

**Strengths:**

- Originality: The integration of hyperbolic geometry with multi-modal structural representations (global structure + active site graphs) for enzyme classification is novel and elegant.
- The hierarchical nature of EC numbers fits naturally with hyperbolic geometry.
-  Combining sequence, structure, and active site information offers richer functional understanding than sequence-only baselines.
- Strong empirical performance: PoinnCARE consistently outperforms 12 state-of-the-art methods (ProtT5, ESM-2, CLEAN, etc.), especially on low-similarity and multi-functional enzyme subsets.
- Hyperbolic representations are robust and efficient: achieve comparable or better accuracy at much smaller dimensions (e.g., 32 vs. 512) compared to CLEAN representations.
Expanded the CARE dataset with active sites and structure for the protein-ml community.

**Weaknesses:**

Presentation clarity:
- While well-written, the methods section could better explain the intuition behind “graph diffusion” and “dual hyperbolic alignment” for readers unfamiliar with those concepts.

-Requiring active site motif limits generalization to unknown enzyme reactions or reactions that lack detailed enzymology studies.

**Questions:**

- How sensitive is the performance to the choice of hyperbolic dimension or curvature parameter?
- How does the model handle noisy or missing active-site data?
- What is the training time and computational cost compared to CLEAN?

---

> ### Author Response · Authors · 2025-11-21
>
> We thank the reviewer for the effort in reviewing our work. We truly appreciate the thoughtful comments and constructive feedback. Our point-to-point responses are as follows.
>
> ---
>
> **RW1.** Thank you for bringing up this concern. We will polish our manuscript and provide more explanations to enhance the readability.
>
> ---
>
> **RW2 & RQ2.** For unannotated enzymes, active sites can be automatically inferred via Folddisco. Specifically, Folddisco retrieves known active site motifs from the training set on the surfaces of unannotated protein structures by constructing an index over position-independent geometric features, and therefore does not rely on detailed enzymological data provided by the user.
>
> We do acknowledge, however, that generalization is limited for rare reactions whose interaction motifs are entirely absent from the training set. This is a common challenge faced by current automatic enzyme function annotation approaches. Addressing these unseen mechanisms and extending predictions to substrates remains a key focus of our future work.
>
> ---
>
> **RQ1.** We evaluated the performance of PoinnCARE across various dimensions, ranging from 512 to 32, with results presented in Figure 5 of our manuscript. Compared to the strong baseline CLEAN, PoinnCARE consistently demonstrates robust performance at all tested dimensions, especially in the highly compact setting with a dimension of 32, where CLEAN decreases to an accuracy of 0.354 at level 4, while our PoinnCARE maintains an accuracy of 0.606 on the <30 Identity test set.
>
> Additionally, we assessed the effect of varying the hyperbolic curvature $c$, including fixed values of $c=-0.5, -1, -5$ as well as a learnable $c$, as detailed in Table 18 in our manuscript. Comparable performance was observed across these settings, with the better results achieved at $c=-1$, making it a simple and practical choice.
>
> ---
>
> **RQ3.** All the experiments are conducted on a single GPU with 40GB of memory. PoinnCARE completed its training and inference in 36 minutes, while CLEAN required approximately 6 hours for training alone. Although CLEAN employs a relatively simple neural network architecture, its contrastive learning framework requires computing all pairwise distances for negative sampling, which substantially increases the overall training time.
>
> ---
>
> Thank you again for your thoughtful feedback and support. We are happy to respond to any further questions.

---

### Author Response · Authors · 2025-11-21
**Summary of Manuscript Revisions**

We sincerely thank the reviewers for their time and constructive feedback. In response, we have revised our manuscript, with changes highlighted in *blue*. The main revisions are summarized below:

- Expanded the related work discussion in Section 2.
- Added a Venn diagram illustrating the relationship distribution between the two similarity graphs in Appendix B.3.
- Provided a case study demonstrating how active sites offer complementary and accurate insights in Appendix E.
- Analyzed misclassification patterns in Appendix F.

We appreciate the reviewers’ valuable comments again and are happy to address any further questions or concerns.

---

### Author Response · Authors · 2025-12-02
**Rebuttal overview before the leakage**

Dear PC, SAC, and AC:

We acknowledge the current situation and appreciate the proposed course of action outlined by ICLR. We are sincerely grateful for your time, effort, and careful coordination throughout the discussion process. To ease the evaluation of our work, we provide an overview of the rebuttal submitted before the OpenReview leakage for your reference.


**Reviewer ZnFn: Initial score of 8.**

**Reviewer e3qB: Initial score of 8. The reviewer acknowledged the discussion and maintained the score (48 hours before OpenReview leakage).**

**Reviewer B891: Initial score of 4. The reviewer confirmed that concerns were addressed and increased the score to 6 (10 hours before OpenReview leakage).**

Additionally, we summarize our discussion with Reviewer **B891** as follows for your reference.

- In **W1&Q1**, the reviewer pointed out the lack of comparison with advanced models such as EnzymeCAGE and GraphEC. **In response**, we clarified the fundamental differences between our work and EnzymeCAGE and included a comparison with GraphEC, demonstrating that PoinnCARE outperforms GraphEC across all metrics (Table r1).

- In **W2&Q2**, the reviewer questioned performing GNN operations in the origin's tangent space rather than each node's local tangent space. **In response**, we clarified that origin-based tangent aggregation is widely adopted in prior work for its simplicity and efficiency, while center-node-based aggregation introduces significantly higher complexity with only limited improvements, as acknowledged by HGCN.

- In **W3&Q3**, the reviewer questioned the contribution of the active site modality. **In response**, we provided *(1)* detailed ablation studies (Table r2), *(2)* a representative case study, and *(3)* a Venn diagram comparing structural and active site graphs, demonstrating the effectiveness and uniqueness of the active site modality.

- In **W4&Q4**, the reviewer questioned the simple weighted sum fusion strategy. **In response**, we compared it against concatenation-based fusion (Table r3), showing that weighted sum achieves competitive or superior performance across test sets.

- **In response to W5, W6&Q6**, we revised our manuscript and included **enhanced related work** and a **comprehansive analysis on failure samples.**

- **In response to Q5**, we clarified the threshold selection protocol and supplemented the performance results under varying thresholds (Table r4 & Table r5).

- **In response to Q7**, we provided the requested running time analysis.

- **In response to the follow-up question**, we provided the Euclidean performance under increasing dimensions (Table r6), further demonstrating the effectiveness and efficiency of the hyperbolic geometry.

We believe our comprehensive responses have adequately addressed all major concerns raised by reviewers, as reflected in the improved and maintained positive scores. We sincerely thank all reviewers for their constructive feedback. We would like to express our deep gratitude again to the PC, SAC, and AC for the time and dedicated effort in coordinating a fair evaluation process under these challenging circumstances.


Warm regards,

Authors

---

### Meta-Review · Area_Chair_dLxB · 2026-01-16

**Summary:**

This paper introduces a method for enzyme function classification. The introduced method called PoinnCARE is multi-modal and uses sequences, structure, and active site motifs into one representation.
These embeddings are used to train GNNs for Enzyme Commission (EC) classification. This is an interesting paper with novel ideas that are well-executed.

**Reviewer Concerns:**

All the reviewer concerns were sufficiently addressed by the authors.

**Reviewer Scores:**

n/a

---

### Decision · Program_Chairs · 2026-01-26

Accept (Poster)